# Intrinsic functional neuron-type selectivity of transcranial focused ultrasound neuromodulation

Kai Yu [1,2,4], Xiaodan Niu[1,4], Esther Krook-Magnuson [3] & Bin He [1✉]

Transcranial focused ultrasound (tFUS) is a promising neuromodulation technique, but its mechanisms remain unclear. We hypothesize that if tFUS parameters exhibit distinct modulation effects in different neuron populations, then the mechanism can be understood through identifying unique features in these neuron populations. In this work, we investigate the effect of tFUS stimulation on different functional neuron types in in vivo anesthetized rodent brains. Single neuron recordings were separated into regular-spiking and fast-spiking units based on their extracellular spike shapes acquired through intracranial electrophysiological recordings, and further validated in transgenic optogenetic mice models of light-excitable excitatory and inhibitory neurons. We show that excitatory and inhibitory neurons are intrinsically different in response to ultrasound pulse repetition frequency (PRF). The results suggest that we can preferentially target specific neuron types noninvasively by tuning the tFUS PRF. Chemically deafened rats and genetically deafened mice were further tested for validating the directly local neural effects induced by tFUS without potential auditory confounds.

[1] Department of Biomedical Engineering, Carnegie Mellon University, Pittsburgh, PA, USA. [2] Department of Biomedical Engineering, University of Minnesota, Minneapolis, MN, USA. [3] Department of Neuroscience, University of Minnesota, Minneapolis, MN, USA. [4]These authors contributed equally: Kai Yu, Xiaodan Niu. ✉email: bhe1@andrew.cmu.edu

Neuromodulation delivers controlled physical energy and intervenes in the nervous system to treat and improve the quality of life of subjects suffering from neurological disorders. For decades, a myriad of brain neuromodulatory approaches, such as deep brain stimulation[1], transcranial magnetic stimulation (TMS)[2,3], transcranial current stimulation (TCS)[4,5], optogenetics[6–8], designer receptors exclusively activated by designer drugs[9,10], sonogenetics[11,12], microbubble-assisted drug delivery[13] and sonoselective transfection[14], and nanomaterial-mediated magnetic stimulation[15–18] have been developed in order to modulate and study the brain.

As a promising new technique, low-intensity transcranial focused ultrasound (tFUS) can be utilized in many neuromodulation applications due to its high-spatial specificity (compared to TMS and TCS[19]) and its non-invasive nature[20]. During tFUS neuromodulation, pulsed mechanical energy is transmitted through the skull with high-spatial selectivity[21], which can be steered[22] and elicit activation or inhibition through parameter tuning[23,24]. Pilot studies investigated the neural effects of ultrasound parameters, such as ultrasound fundamental frequencies ($f_0$)[24], durations (UD)[23], duty cycles (DC)[25,26], pulse repetition frequencies (PRF)[26], etc. Besides a few human studies[21,27,28], animal models, such as rodents[24,29,30], sheep[31,32], swine[33,34], and monkeys[35–39], were used to investigate the effects of ultrasound neuromodulation. tFUS were observed to induce behavioral changes[23,39,40], electrophysiological responses through, e.g., electromyography[24,41], electroencephalography[30,42], local field potentials (LFPs)[29,43], and multi-unit activities (MUAs)[29] with high in vivo temporal/spatial measurement fidelity, or neurovascular activities, e.g., blood-oxygen-level-dependent signal[36–38,44], etc.

There are multiple theories of the neuromodulatory effect of ultrasonic waves on neurons[45], ranging from lipid bilayer membranes[25,46,47] to ion channels[26,48]. Since different types of neurons are known to have vastly different ion channel concentrations, morphology and size, cell-type-specific effect of ultrasound is a natural extension of the theory. To achieve selectivity in stimulating brain circuits or even among cellular populations, focused ultrasound was employed in combination with specific-neuromodulatory-drug-laden nanoparticles[49,50], cell-specific expression of ultrasound sensitizing ion channels[11] or acoustically distinct reporter genes in microorganisms[51]. Based on previous studies, it was evident that distinct tFUS parameters can exhibit unique stimulation effects. So far, none has explored the intrinsic effects of the wide range of ultrasound parameters on specific neuron subpopulations, which may pave the way for the translation of tFUS as an effective non-invasive modulation tool for brain conditioning and facilitating the elucidation of neural mechanisms. Previous works in a theoretical model[25] and in vitro experiments[52] showed feasibility of cell-type-specific effects in directly stimulating neurons; however, the highly needed in vivo evidence of cell-type-specific effects of tFUS has not been shown yet.

In this work, we investigate the functional cell-type dependent effects of tFUS stimulation through extracellular recordings in in vivo rodent brains. Leveraging on distinct features in extracellular action potential waveforms in different functional cell-types, we analyze the unique neuronal responses of two functional cell-types under various tFUS stimulation profiles with different ultrasound PRFs.

## Results

### Characterizing tFUS stimulation and setup.
The in vivo experimental setup and ultrasound temporal waveforms are illustrated in Fig. 1. As presented (Fig. 1a), tFUS was targeted over the left somatosensory cortex at an incidence angle of 40°[53]. The stimulation dynamics of the tFUS waveforms consisted of

**Table 1 Administered tFUS conditions with constant TBD.**

| tFUS conditions[a] | PRF (Hz) | DC (%) | $I_{SPTA}$ (mW/cm$^2$) |
|---|---|---|---|
| PRF 30 Hz | 30 | 0.6 | 1.05 |
| PRF 300 Hz | 300 | 6 | 10.55 |
| PRF 1500 Hz | 1,500 | 30 | 52.74 |
| PRF 3000 Hz | 3,000 | 60 | 105.48 |
| PRF 4500 Hz | 4,500 | 90 | 158.22 |

[a]Except for the ultrasound pulse repetition frequency (PRF), the ultrasound duty cycle (DC), and the spatial-peak temporal-average intensity ($I_{SPTA}$), all of the listed tFUS conditions keep the following parameters constant, i.e., ultrasound fundamental frequency ($f_0$ = 500 kHz), ultrasound duration (UD = 67 ms), cycle per pulse number (CPP = 100), tone-burst duration (TBD = 200 μs), spatial-peak pulse-average intensity ($I_{SPPA}$ = 175.80 mW/cm$^2$), and spatial-peak temporal-peak intensity ($I_{SPTP}$ = 351.61 mW/cm$^2$). Note: the inter-sonication interval (ISoI) has a typical value of 2.5 s and has 10% jittering. This 10% jittering of the inter-stimulus interval is to minimize the timing effect and potential brain adaptation to the tFUS stimulation.

tone-burst sinusoidal waves with a constant $f_0$ (500 kHz) and tone-burst duration (TBD, 200 μs), and varied PRF, spanning five levels between 30 and 4500 Hz (Fig. 1b, Table 1 in Methods). 32-channel micro-electrode array was inserted at S1 (i.e., primary somatosensory cortex S1, ML: −3 mm, AP: −0.84 mm, depth: 1 mm, Fig. 1c, d). The inter-sonication interval (ISoI) was 2.5 s per trial (10% jittering for some ISoI).

To control for possible confounds due to acoustic and electromagnetic noise in the experimental setup, three sham ultrasound conditions were conducted. The sham condition, sham with flipped transducer (SFLP), transmitted ultrasound waves in air, directed 180 degrees away from the skull (Fig. 1e). The second sham condition, sham at skull front (SSKF), in which tFUS was directed at a secondary control site physically away from the target site, was used to control for the skull bone conduction (Fig. 1f). As we set out to record the neuronal activities at the tFUS-targeted brain using a silicon-based microelectrode array (MEA), we investigated whether the mechanical interaction between the ultrasound field and the silicon electrode shank may induce confounding neuronal responses. Hence, a third sham condition, sham at electrode shank (SSHK), was designed applying ultrasound directly to the upper part of the electrode shank (Fig. 1g, see below for detailed demonstration) in order to produce multiple levels of mechanical vibration at the electrode tip as in the experimental conditions directly applying tFUS at S1.

A hydrophone-based three-dimensional (3D) ultrasound pressure scanning system was used to experimentally obtain measurements of the tFUS spatial profile behind a freshly excised rat cranium. To illustrate the tFUS spatial distribution, the spatial profile of the transcranial ultrasound pressure in the X–Y plane measured from ex vivo measurement was superimposed on a rat cranium (Fig. 2a). The ultrasound spatial map from the sagittal view (X–Z plane) was reconstructed, in which the mechanical energy was observed to be distributed along a sagittal beam up to a depth of 5 mm (−3 dB along the axial direction), but spatial-peak pressure (i.e., 79 kPa) was focused within 1 mm behind the skull (Fig. 2b). The ultrasound field without the presence of cranial bone was scanned with a spatial-peak pressure of 148 kPa (Fig. 2c). This illustrates that the 40-degree angled incidence[53] led to a shallow targeting at the rat cortex. Angled tFUS stimulation was the preferred stimulation configuration for studying the cortex, as its activation pattern is shallower than normal incident tFUS. Numerical simulations were conducted using a pseudo-spectral time domain method (see Methods) of spatial distributions of the ultrasound pressure field throughout the imaged rat skull in order to account for potential acoustic reflections from the skull base in the small enclosed cavity (Fig. 2d, e, with extensive numerical simulations in Supplementary Fig. 1).

During tFUS stimulation, increased time-locked neuronal firing was observed in recorded MUA as compared to SFLP,

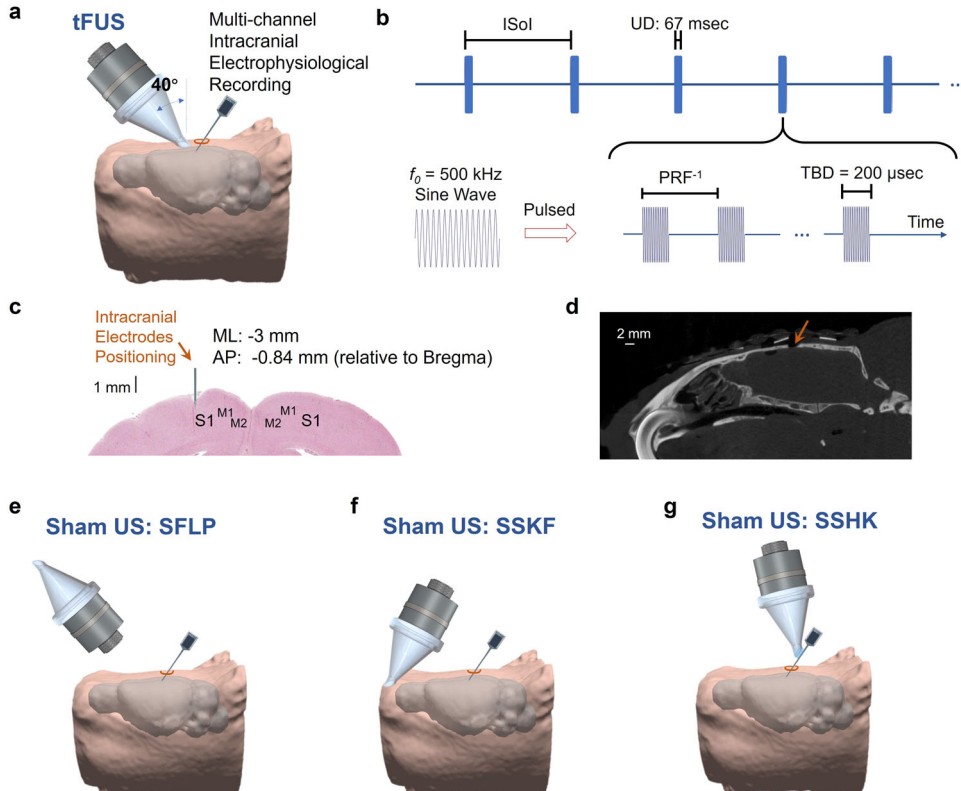

**Fig. 1 The experimental setup of ultrasound stimulation and intracranial electrophysiological recording. a** Collimator guiding focused ultrasound to hair-removed scalp of an anesthetized rat model with an incidence angle of 40°, and a 32-channel electrode array inserted at an incidence angle of 40° into the left primary somatosensory cortex (S1) prepared through a craniotomy. The rat head and brain model were 3D reconstructed from $T_2$-weighted (T2W) MRI images[94]. **b** The temporal profile of the transcranial focused ultrasound (tFUS). One-hundred cycles of sinusoidal wave formed a single ultrasound pulse, which generated a tone-burst at an ultrasound fundamental frequency ($f_0$) of 500 kHz for a tone-burst duration (TBD) of 200 μs. Such ultrasound pulses were repeated at certain PRF for a corresponding number within the ultrasound duration (UD) of 67 ms. The inter-sonication interval (ISoI) was 2.5 s, and some ISoI had 10% jittering, in which the ISoI was randomly selected from a uniform distribution centered at 2.5 s, bounded by ±250 ms. **c** The spatial coordinates of electrophysiological recordings. The coronal brain slice shows the location of electrode insertion with the insertion depth of 1 mm at the left S1. ML denotes the medial lateral distance from midline; AP denotes the anterior posterior distance from Bregma. **d** A sagittal view of the micro-CT image captured the surgical burr hole (approximate diameter: 2 mm) on the top of cranium. The micro-CT scan was repeated for six times independently with similar results. **e** The experimental setup of the sham ultrasound condition as a negative control (SFLP). The acoustic aperture of the ultrasound transducer was flipped by 180°, while the pulsed ultrasound and the intracranial recordings were maintained. **f** The experimental setup of another sham ultrasound condition as a control for bone-conduction (SSKF). The ultrasound incidence took place at an anterior location of the skull with active ultrasound transmission. **g** Another sham control on electrode shank, i.e., SSHK, is delivered to investigate potential effects of electrode vibrations on neuronal activation. As shown, in the SSHK sham condition, ultrasound is delivered to the shank of the electrode away from the recording sites as further depicted in Fig. 5d.

SSKF and SSHK conditions (Fig. 3a). Furthermore, the spatial specificity of the ultrasound-induced brain activities was demonstrated with neural recording at the primary auditory cortex (A1) when tFUS was targeted at S1 (Supplementary Fig. 2a–c), and a comparison of the recorded signal was also conducted in the same animal before and after euthanasia to illustrate any tFUS artifacts in the recording system (Supplementary Fig. 2d). With necessary preprocessing (see details in Methods), all tFUS aligned activations, such as the aforementioned MUA (Fig. 3a) and LFP (Supplementary Fig. 3), observed at the electrodes were deemed to be direct neural responses to the ultrasound energy depositions.

**Rat somatosensory cortex suggests functional neuron-type-specific response to tFUS.** All recorded action potentials from the 32-channel electrode array were sorted offline based on the spike waveforms and inter-spike intervals (ISpI). Regular-spiking (presumably excitatory) (RSU) and fast-spiking (presumably

inhibitory) units (FSU) were thus identified based on the temporal dynamics of the action potential waveform[54–58] shown as in Fig. 3b, e. The extracted features were the durations of initial phase (IP) of the action potential, i.e., from onset to the re-crossing of baseline, and afterhyperpolarization period (AHP), i.e., from the end of the IP to its re-crossing of baseline (Fig. 3b, e). The differences in these features have been associated with different types and distributions of ion channels in the neuronal cell membrane. We thus hypothesized that the RSU and FSU have distinct responses to various tFUS stimulation sequences due to their intrinsic cellular differences, such as mechano-sensitive ion channels distribution[52,59] or cell morphology, etc.

After recording the MUAs from the first group of wild-type male Wistar rats ($N = 9$), we studied the neural effects of the administered pulsed tFUS through intracranial MUA recordings. Using peri-stimulus time histograms (PSTH), we found a significant increase of spike rate ($6.23 \pm 1.10$ spikes/s, Fig. 3c) in a regular-spiking somatosensory cortical neuron (mean spike waveform IP: 0.85 ms, AHP: 1.8 ms) when stimulated with a tFUS

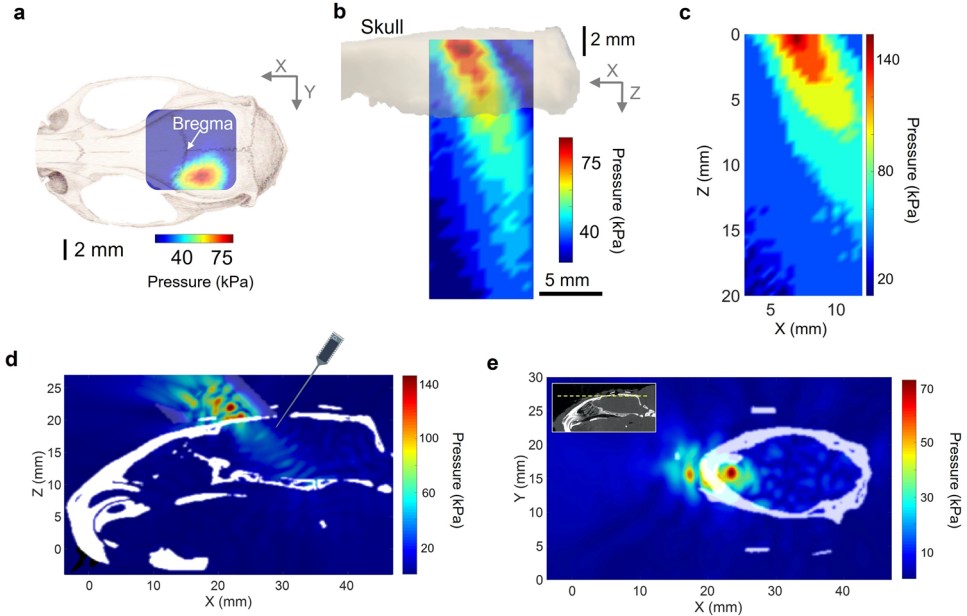

**Fig. 2 The characterizations of the tFUS pressure fields. a, b** One transverse (X–Y plane) and one sagittal (X–Z plane) scans of ultrasound pressure distribution under the cranium using a hydrophone-based US field mapping system. After penetrating through the ex vivo skull top, spatial-peak ultrasound pressure is measured as 79 kPa. The ultrasound intensities are listed in Tables 1 and 2. **c** A sagittal (X–Z plane) view of ultrasound pressure distribution in the free water without the cranial bone. The ultrasound setup, including the use of collimator and incidence angle, were the same as depicted in Fig. 1a. The spatial-peak ultrasound pressure is measured as 148 kPa. **d** A 3D computer simulation investigating the quantitative maximum pressure distribution at one sagittal i.e., X–Z plane inside the rat skull (in white) when the tFUS is guided by a 3D-printed collimator (see Methods) and directed to the S1 with an incidence angle of 40°. As illustrated, the intracranial electrode array is inserted into the brain via the cranial burr hole. **e** A transverse (X–Y plane) view of computer simulated ultrasound pressure field. The yellow dashed line shown in the inset indicates the location of the presented transverse plane.

condition (PRF = 300 Hz, see ultrasound intensities in Table 1), and a further increased spiking rate (14.35 ± 1.65 spikes/s, Fig. 3d) was observed in response to the increased sonication temporal energy (PRF = 3000 Hz, see Table 1 for ultrasound intensities). For a more intuitive comparison, increases of spiking as a function of time along 478 consecutive trials were demonstrated with the raster plot (Fig. 3c, d), in which the density of spiking events increased during the ultrasound stimulation. For another identified RSU, the comparisons were shown between the tFUS and sham conditions (Supplementary Fig. 4). Using the PRF of 1500 Hz, a significant bursting peak, a type of non-Poissonian, tFUS-mediated response was shown in the time histogram (Supplementary Fig. 4a) and in the return plot of ISpI (Supplementary Fig. 4b). In contrast, the SFLP and SSKF conditions did not exhibit such distinct spiking increase due to stimulation seen from either PSTH/raster (Supplementary Fig. 4c, e) or return plots (Supplementary Fig. 4d, f).

On the contrary, a fast-spiking cortical neuron (Fig. 3e) with shorter durations of IP (mean: 0.70 ms) and AHP (mean: 0.65 ms) demonstrated a more homogeneous PSTH distribution in response to the levels of tFUS treatment (e.g., PRF 300 Hz and PRF 3000 Hz in Fig. 3f, g). The return plots (Supplementary Fig. 5b, c) illustrated a fast-spiking behavior with a shorter refractory period than the RSU identified in Fig. 3b. As seen in the example (Fig. 3f), FSU firing rates were not significantly disturbed by tFUS, i.e., that the firing rate (7.60 ± 1.20 spikes/s) was not significantly altered by the ultrasound stimulation (PRF = 300 Hz) comparing to pre-stimulus rates (e.g., 7.50 ± 1.20 spikes/s at the time bin of [−0.05, 0] s). Interestingly, for FSUs, no significant changes in spike rates (6.50 ± 1.30 spikes/s) was observed even when ultrasound was administered at a PRF of 10 times higher (PRF = 3000 Hz, Fig. 3g). Furthermore, the return plots (Supplementary Fig. 5b, c) indicated that although this fast-spiking neuron did not significantly change its rate of

firing action potentials in response to tFUS, there was a possible trend of changing the spiking pattern of its bursting mode.

Given these case studies, we pursued a statistical investigation of the behavior of different neuron types in response to tFUS and sham conditions, across multiple levels of PRFs. We compiled all single unit activities during the sonication period (i.e., UD) of 67 ms recorded from the rats and separated them into RSUs and FSUs using k-means cluster analysis (Fig. 4). In the tFUS group, 199 identified single units were separated into the two groups, with the RSU group containing 146 units (Fig. 4a). The sample sizes were unbalanced due to the different prevalence of each cell type in the cortex. This unbalance between FSU and RSU numbers was mainly observed under ketamine/xylazine cocktail anesthesia, and under isoflurane anesthesia, such an unbalance was improved. The baseline firing rate of each single unit in different ultrasound sessions, which was affected by anesthetic depth with ketamine/xylazine cocktail and time duration of recordings, was mitigated by randomizing the order of ultrasound conditions on each animal subject for rigorous statistical comparisons.

In the RSU group, we found PRF levels have a statistically significant effect on the neuron firing rates (Kruskal–Wallis chi-squared = 14.45, p = 0.006), indicating that the RSUs respond to different tFUS conditions with different firing rates (Fig. 4b). More specifically, the spiking rate increases also showed a positive correlation with the increasing PRFs. Through the multi-comparisons, we found that in an anesthetized rat model, the RSUs significantly increase their firing rates in response to PRFs at 3000 and 4500 Hz when both comparing to that induced by a low PRF at 30 Hz (Fig. 4b, PRF 30 Hz vs. PRF 3000 Hz: p = 0.003; PRF 30 Hz vs. PRF 4500 Hz: p = 0.0004). Whereas in the FSU group (Fig. 4c), no significant difference between tFUS conditions can be found (Kruskal–Wallis chi-squared = 4.34, p > 0.3). This implied that the spike rates of the FSUs were not

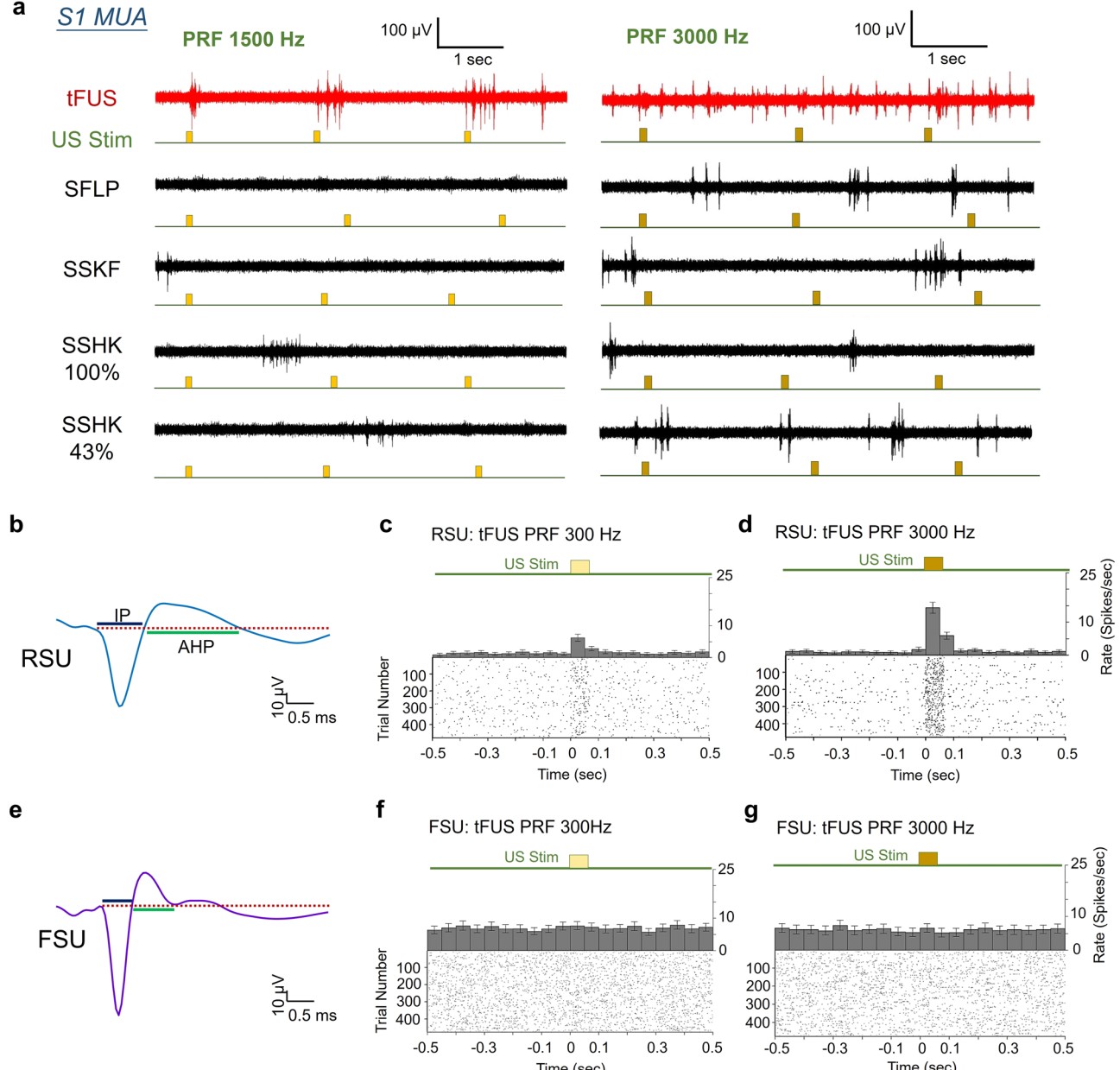

**Fig. 3 The neuronal spiking activities in response to tFUS. a** Examples of acquired multi-unit activity (MUA) from the S1 using tFUS with PRF = 1500 Hz (left) and PRF = 3000 Hz (right) show time-locked activity during tFUS stimulation. The timing of the ultrasound-induced action potentials with respects to administered stimulations (the timing of ultrasound stimulation is added with 10% jittering presented below each MUA trace) are exemplified with three trials. The sham conditions, i.e., SFLP, SSKF, SSHK 100% and SSHK 43% (see Methods for descriptions of SSHK levels) show disappearance of time-locked MUAs. These MUAs are obtained from the in vivo rats anesthetized with isoflurane. **b** A typical example of a regular-spiking unit (RSU) separated from the recorded MUAs. The waveform features, i.e., time durations of initial phase (IP) and afterhyperpolarization (AHP) were extracted to conduct the units' separation. **c, d** Representative peri-stimulus time histograms (PSTHs, bin size: 50 ms, $n = 478$ trials for each time bin) and raster plots of one RSU (IP duration mean: 850 μs; AHP duration mean: 1850 μs) responding to the PRF of 300 Hz (**c**) and 3000 Hz (**d**). **e** An example of a fast-spiking unit (FSU) identified with short IP and AHP durations. **f, g** Representative PSTHs (bin size: 50 ms, $n = 478$ trials for each time bin) and spiking raster plots of one FSU (IP duration mean: 700 μs; AHP duration mean: 600 μs) in response to the two tFUS conditions with PRF of 300 Hz (**f**) and 3000 Hz (**g**), respectively. The time histograms and return plots of the inter-spike interval (ISpI) are depicted in Supplementary Figs. 4 and 5. All of the temporal dynamics were computed across 478 trials. The applied ultrasound conditions are described in Table 1 (see Methods). Data are shown as the mean ± 95% confidence interval in the PSTHs. **c, d, f, g** are example PSTHs from the same subject, same unit under tFUS stimulation at 300 Hz and 3000 Hz PRF. Source data are provided in the Source Data file.

significantly altered by the levels of PRF, consistent with the case studies (Fig. 3f, g and Supplementary Fig. 5a). Furthermore, the interaction between the neuronal cell type, i.e., RSU/FSU, and the PRF levels was further investigated using a two-way ANOVA. Although this interaction was not statistically significant ($p > 0.5$) when the rats were anesthetized with the ketamine and xylazine, the PRF was still deemed as a significant factor ($p = 0.042$).

The contrast between the responses observed in these two different neuron types suggested a functional cell-type-selective

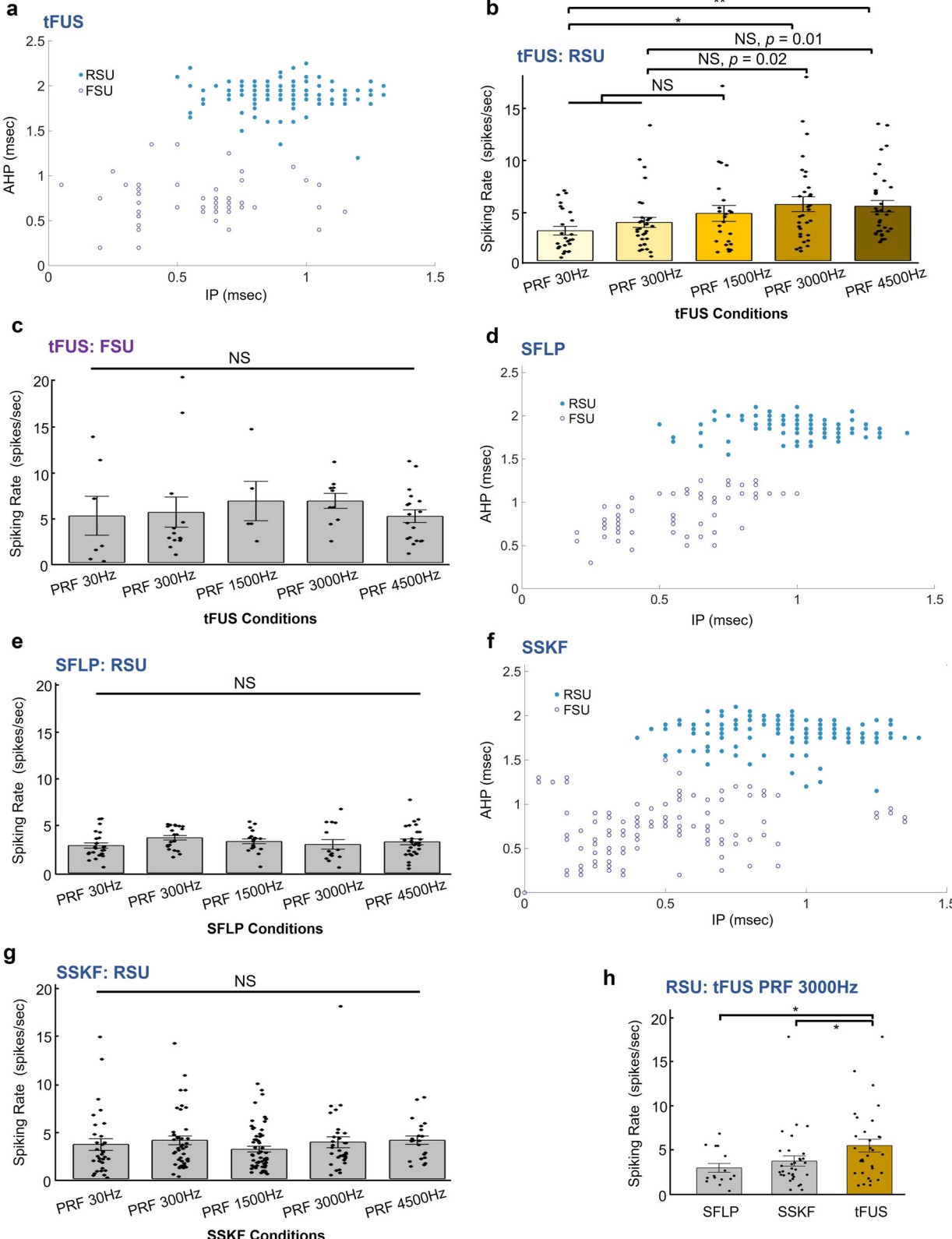

mechanism by tFUS. Unsurprisingly, the RSU group did not show significant differences among the five levels of sham ultrasound conditions, including the SFLP (Kruskal–Wallis chi-squared = 6.58, $p > 0.15$, Fig. 4e) and SSKF (Kruskal–Wallis chi-squared = 6.32, $p > 0.15$, Fig. 4g) shams. Moreover, as seen in Fig. 4b, the first significant difference shows up at the PRF of 3000 Hz; when the tFUS was compared with the corresponding

sham conditions, significant increase of spiking rates was found in the tFUS vs. SFLP and SSKF at this PRF as well (Fig. 4h, tFUS vs. SFLP: $p = 0.033$; tFUS vs. SSKF: $p = 0.048$).

In order to rule out the possibility that the observed functional cell-type selectivity may originate from the electrode vibration due to the tFUS incidence and maintain a constant anesthetic depth during the neuronal recordings, we further investigated this

**Fig. 4 Cell-type-selective responses to tFUS and sham ultrasound conditions.** The ultrasound parameters listed in Table 1 were used in this study. **a** $k$-means cluster classification of 199 single units with blue solid circles depicting the RSUs while purple circles representing the FSUs. The majority was classified as the RSUs with longer IP and/or AHP durations than the FSUs. These spiking units were recorded and identified under the influence of the administered anesthesia (xylazine) and analgesic drugs (ketamine), in which the durations of both IP and AHP are observed to be longer than the results reported in literature[56,95] for different sedative approaches. **b** The 146 RSUs showed significantly different responses to different tFUS conditions (Kruskal–Wallis chi-squared $= 14.45$, $p = 0.006$). Data are shown as the mean±s.e.m., with statistical comparisons made through Kruskal–Wallis two-sided one-way ANOVA on ranks and post hoc one-tail two-sample Wilcoxon tests with Bonferroni correction for multiple comparisons (PRF 30 Hz vs. PRF 3000 Hz: $p = 0.003$; PRF 30 Hz vs. PRF 4500 Hz: $p = 0.0004$). *$p < 0.005$, **$p < 0.001$. NS, not significant. s.e.m., standard error of mean. **c** The FSU group, which consisted of fewer neurons ($N = 53$, marked as purple hollow circles), was observed to have a higher mean spike rate than RSUs, whereas this FSU group showed no significant effect by different PRF levels. Data are shown as the mean±s.e.m., statistics by Kruskal–Wallis two-sided $H$-test. **d, e** In one of the sham conditions, i.e., SFLP, the cluster analysis for another identified 179 single units. **d** 110 RSUs and 69 FSUs were separated. **e** RSUs ($n = 110$) showed homogeneous lack of response to the SFLP conditions. No significant effect of the ultrasound conditions was observed through statistics by Kruskal–Wallis two-sided $H$-test. Data are shown as the mean ± s.e.m. NS, not significant. s.e.m., standard error of mean. **f, g** In another sham control, i.e., SSKF, for investigating any potential effects of bone-conduction, the cluster analysis for another identified 311 single units. **f** 192 RSUs and 119 FSUs were separated. **g** RSUs ($n = 192$) also showed a homogeneous lack of response to the SSKF sham ultrasound conditions. No significant effect of the ultrasound conditions was observed through statistics by Kruskal–Wallis two-sided $H$-test. NS, not significant. s.e.m., standard error of mean. **h** Group comparisons between SFLP ($n = 15$) and tFUS ($n = 31$), and between SSKF ($n = 32$) and tFUS conditions regarding RSUs' spiking rates responding to the PRF 3000 Hz level (tFUS vs. SFLP: $p = 0.033$; tFUS vs. SSKF: $p = 0.048$). Data are shown as the mean ± s.e.m., statistics by two-tail Wilcoxon test. *$p < 0.05$. Source data are provided in the Source Data file.

cell-type selectivity in another batch of rats ($N = 10$) anesthetized with isoflurane. The advantage of using isoflurane over ketamine/xylazine cocktail was the ability to maintain relatively constant levels of anesthetic effects across experiment trials. The functional cell-type selectiveness between RSU and FSU groups was only presented when tFUS directly interacts with the cortical brain. For 245 identified RSUs, they demonstrated a significant difference in response to the five levels of PRF with normalized spiking rates (Fig. 5a, Kruskal–Wallis chi-squared $= 42.02$, $p = 1.65 \times 10^{-8}$); while the clustered 347 FSUs were not sensitive to the PRF change (Kruskal–Wallis chi-squared $= 6.16$, $p = 0.19$, Fig. 5b). As expected, a much significant interaction ($p = 4.09 \times 10^{-7}$) between the cell type and the PRF levels was observed in a two-way ANOVA model (Fig. 5c). Moreover, once the anesthesia method was incorporated as the third factor for the ANOVA, the three-way interaction among cell type, PRF levels, and anesthesia method (i.e., using ketamine/xylazine cocktail with normalizing the data presented in Fig. 4a–c or isoflurane with the normalized data presented in Fig. 5a–b) is statistically significant ($p = 3.37 \times 10^{-9}$). This indicates that the cell type and PRF interaction varies significantly across different anesthesia methods. Another 2-way ANOVA studying how the RSUs' activities were impacted by both the PRF levels and anesthesia methods was illustrated in Supplementary Fig. 6i. Significant effects of the PRF ($p = 1.97 \times 10^{-6}$) and the anesthesia method ($p = 0.0069$) were found.

Further, applying ultrasound to the electrode shank (Fig. 5d) with the same transmitted ultrasound energy during the recordings, the 100% SSHK conditions (see details of SSHK levels in Methods) did not show significant spiking rate change for RSUs and FSUs (Kruskal–Wallis chi-squared $= 1.99$, $p = 0.74$, in Fig. 5e and Kruskal–Wallis chi-squared $= 8.39$, $p = 0.078$, in Fig. 5f, respectively). Another three adjusted levels of ultrasound energy, i.e., 43% SSHK (Fig. 5g, h), 28% SSHK (Supplementary Fig. 6e, f) and 13% SSHK (Supplementary Fig. 6g, h) were further administered, but none of these conditions led the RSUs or FSUs to exhibit significant selectiveness on PRFs. This study demonstrated that the functional cell-type selectivity of tFUS derives from ultrasound-brain interaction but not from the mechanical vibrations at the recording electrodes.

**Functional neuron-type-specific responses to PRF in constant duty cycle.** In the studies presented above, TBD were maintained constant across different PRF levels. This ensures the each tFUS pulse delivers the same ultrasound energy per pulse. However, by

**Table 2 Administered tFUS conditions with a constant DC of 60%.**

| tFUS conditions[a] | PRF (Hz) | TBD (ms) |
|---|---|---|
| PRF 30 Hz DC 60% | 30 | 20 |
| PRF 300 Hz DC 60% | 300 | 2 |
| PRF 1500 Hz DC 60% | 1,500 | 0.4 |
| PRF 3000 Hz DC 60% | 3,000 | 0.2 |
| PRF 4500 Hz DC 60% | 4,500 | 0.134 |

[a]Except for the PRF, CPP and TBD, all of the listed tFUS conditions keep the following parameters constant, i.e., $f_0 = 500$ kHz, UD $= 67$ ms, $I_{SPTA} = 105.48$ mW/cm$^2$, $I_{SPPA} = 175.80$ mW/cm$^2$, and $I_{SPTP} = 351.61$ mW/cm$^2$. Note: the ISoI has a typical value of 2.5 s and has 10% jittering.

doing so, the ultrasound duty cycle, hence $I_{SPTA}$, changes across different PRF levels. In order to further examine whether the observed effects are due to the change of PRF levels or due to the change of DC, we performed an additional set of experiments. In this experiment, we applied tFUS stimulation at the same five PRF levels, while maintaining the same duty cycle, thus $I_{SPTA}$. We hypothesized that functional neuron-type-specific responses to PRF is preserved when the DC is maintained constant.

To keep the $I_{SPTA}$ constant, we changed the TBD associated with each PRF (examples are demonstrated in Fig. 6a for PRF 1500 Hz DC 60% and PRF 3000 Hz DC 60%). The complete ultrasound parameters are listed in Table 2 (see Methods). We delivered such tFUS parameter set and tested on seven rats, under isoflurane anesthesia. All other experimental conditions are maintained. As seen in Fig. 6, when stimulating at a constant 60% duty cycle, different PRFs resulted in significantly different spiking rates of RSUs with Bonferroni correction for multiple comparisons (Fig. 6b, Kruskal–Wallis chi-squared $= 70.61$, $p = 1.69 \times 10^{-14}$), while not significantly changing the spiking rate of FSUs (Fig. 6c, Kruskal–Wallis chi-squared $= 7.84$, $p > 0.09$). As shown in Fig. 6d, the two-way ANOVA presented a significant interaction between the cell type and PRF ($p = 4.51 \times 10^{-10}$). No significant difference was observed under the SSKF conditions (Fig. 6e, f).

Overall, the above results illustrate that using the same 60% DC, PRF can still achieve the cell-type-specific responses between RSUs and FSUs. The consistent observation of functional neuron-type-specific response to different PRFs under both different DCs (the same TBD) and the same DC (different TBDs), suggests that

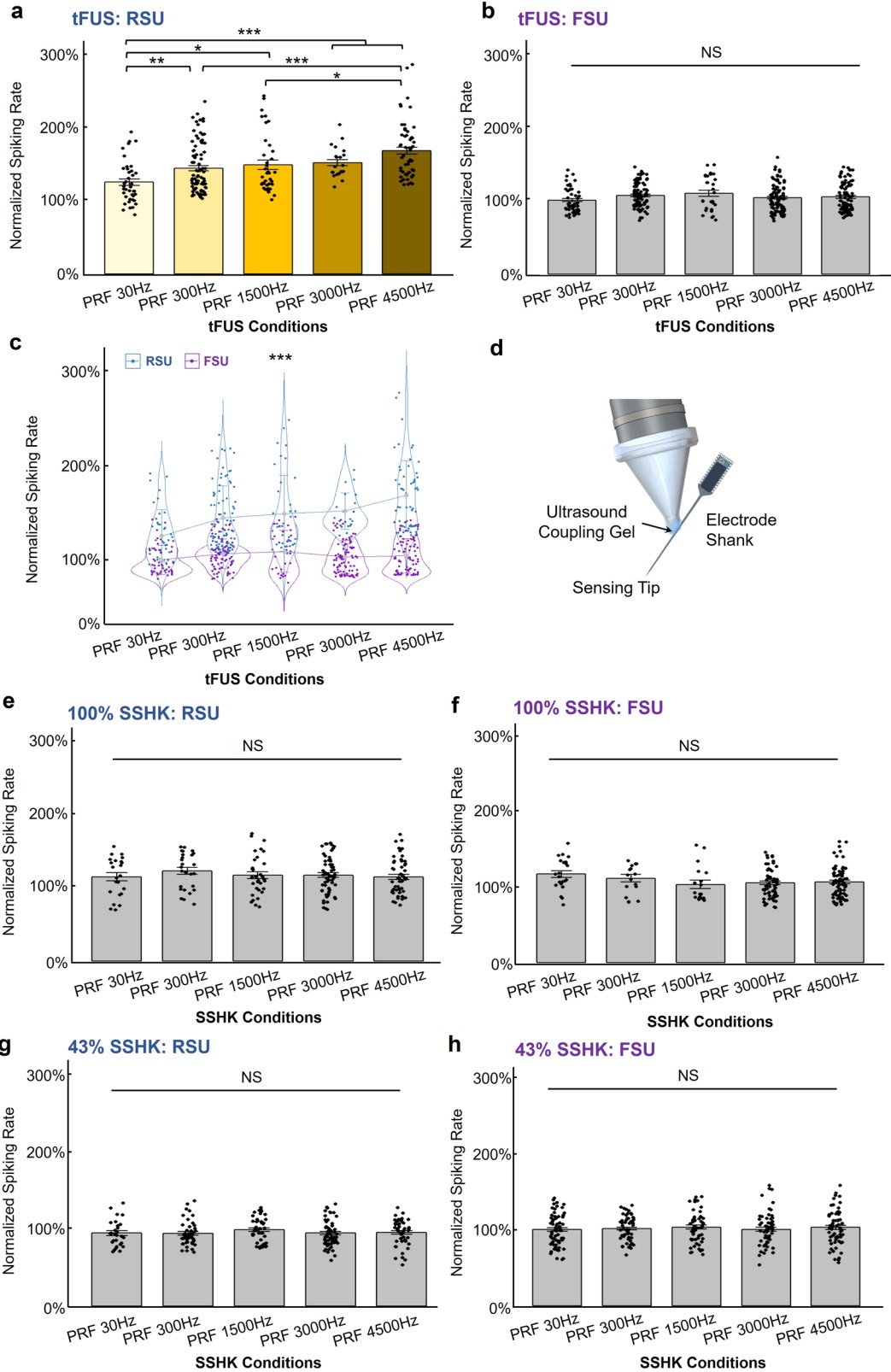

the PRF is the contributing factor to the unequal responses of RSUs and FSUs to the tFUS stimulation. It should be noted that, we do not have direct evidence to refute whether there exists an interaction between PRF and duty cycle, and whether changing both will induce greater changes between RSUs and FSUs. We performed a two-way ANOVA testing factors of PRF levels and

whether TBD is held constant or not, i.e., whether the DC is held constant or not, as well as their interactions. The test was performed between the datasets in Fig. 5a, b and Fig. 6b, c for both RSU and FSU groups, respectively. The data from Fig. 5 used a consistent TBD, thus inconsistent duty cycle across PRFs. Conversely, the data from Fig. 6 used different TBDs, in order to

**Fig. 5 Cell-type-selective responses to tFUS but not to the SSHK sham condition.** The ultrasound parameters listed in Table 1 were used in this study. Isoflurane was used to anesthetize the rats. **a**, **b** The comparison for identified 592 single units across different PRF conditions under direct tFUS stimulation condition with isoflurane anesthesia. 245 RSUs and 347 FSUs were separated. During tFUS condition, RSUs significantly differed their responses to different PRF conditions (Kruskal–Wallis chi-squared = 42.02, $p = 1.65 \times 10^{-8}$). Data are shown as the mean±s.e.m. of normalized firing rates, with statistical comparisons made through Kruskal–Wallis one-way ANOVA on ranks and post hoc one-tail two-sample Wilcoxon tests with Bonferroni correction for multiple comparisons (PRF 30 Hz vs. PRF 300 Hz: $p = 8.42 \times 10^{-4}$; PRF 30 Hz vs. PRF 1500 Hz: $p = 1.76 \times 10^{-3}$; PRF 30 Hz vs. PRF 3000 Hz: $p = 4.84 \times 10^{-5}$; PRF 30 Hz vs. PRF 4500 Hz: $p = 8.36 \times 10^{-9}$; PRF 300 Hz vs. PRF 4500 Hz: $p = 8.43 \times 10^{-6}$; PRF 1500 Hz vs. PRF 4500 Hz: $p = 1.25 \times 10^{-3}$). *$p < 0.005$, **$p < 0.001$, ***$p < 0.0001$. NS, not significant. s.e.m., standard error of mean. **c** In a 2-way ANOVA analysis, a statistically significant interaction ($p = 4.09 \times 10^{-7}$) between the neuronal type, i.e., RSU/FSU, and the PRF levels was found. Data are shown as the mean ±s.d. of normalized firing rates. ***$p < 0.001$. s.d., standard deviation. **d** In the SSHK condition, ultrasound was delivered onto the shank of the electrode base from the recording sites as further depicted. The ultrasound was transmitted onto the electrode shank via ultrasound coupling gel. **e**–**h** Ultrasound levels during SSHK were delivered in two grades 100% (**e**, **f**) and 43% (**g**, **h**). No significant effect was observed in both RSUs and FSUs for all SSHK conditions through statistics by Kruskal–Wallis H-test. NS, not significant. Results from more SSHK levels can be found in Supplementary Fig. 6e–h. See more details about the levels of SSHK conditions in Methods. Data are shown as the mean ± s.e.m. of normalized firing rates, with statistical comparisons made through Kruskal–Wallis one-way ANOVA on ranks with Bonferroni correction for multiple comparisons. s.e.m., standard error of mean. NS, not significant. **e**, **f** The comparison for identified 386 single units across different PRF conditions under direct 100% power delivered at electrode shank stimulation condition. 192 RSUs and 194 FSUs were separated. **g**, **h** The comparison for identified 525 single units across different PRF conditions under direct 43% power delivered onto the electrode shank. 225 RSUs and 300 FSUs were identified and separated. Source data are provided in the Source Data file.

maintain the same duty cycle among PRF levels. In the RSU group, we found a statistically significant different normalized spiking rate effected by both PRF levels ($p < 0.001$) and by whether TBD was held constant or not ($p < 0.001$), though the interaction between these two factors was not significant. The results suggest that changing the TBD could have an effect on spiking rate of RSUs, and maintaining the same duty cycle does not affect the ability of PRF to change the neuronal activities. In contrast, for the FSU group, we did not observe significantly different normalized spiking rate effected by the PRF level ($p > 0.20$) and observed a significant effect by whether TBD was held constant or not ($p = 7.62 \times 10^{-6}$); the interaction between the PRF and TBD was marginally significant ($p = 0.05$).

**Optotagging in mice also suggests functional neuron-type-specific response.** Based upon the observations of intrinsic, unequal responses to the PRF change by two neuronal populations distinct upon discharge patterns, we tested the functional neuron-type specificity hypothesis in transgenic mouse models with parvalbumin (PV, $N = 53$, from 3 mice) and CaMKII-alpha ($N = 50$, from 2 mice) cortical neurons identified by response to optical stimulation. Optogenetics is used to identify excitatory and inhibitory neuron populations, by coexpressing channelrhodopsin only in the PV and CaMKII-alpha expressing neurons. Figure 7a illustrated the in vivo experimental setup combining optical stimulation, tFUS and multi-channel intracranial electrophysiological recordings. The optogenetic stimulation (wavelength = 465 nm) locally activated a subpopulation of channelrhodopsin expressing neurons; based on the MUA recording from optrode, we thus identified the cell type of our recording neurons (Fig. 7b). The waveforms of action potentials of each neuron were illustrated together with the spike rates when receiving the optical stimuli (Fig. 7b). Optogenetic stimulations were administrated after completing all tFUS recording sessions.

By switching animal models from rats to mice, we changed the outlet size of our collimator for the small mouse brain (see Methods for a different outlet size, i.e., circular area of 5.39 mm²) but we maintained the same tFUS transmission parameters. We were not able to replicate the higher tFUS PRFs, i.e., 1500–4500 Hz, given the observed extensive ultrasound field and thus brain activation[60] in the small mice brain model when administering 500 kHz tFUS. At high PRFs, inevitable standing waves were engendered within the small mice skull cavity (Supplementary Fig. 1g). Between PV and CaMKII-alpha neurons, we observed significant differences in the

action potential waveforms regarding the IP and AHP phase durations, validating our method for separating neuron population in rats (Fig. 7c, $p < 0.05$). In PV neurons, spike rates were higher under tFUS stimulation at PRF of 30 Hz (Fig. 7d, $p < 0.001$), while in CaMKII-alpha neurons, significantly higher spike rates were observed during tFUS stimulation at PRF of 300 Hz (Fig. 7e, $p < 0.01$). The difference in response to tFUS PRFs across these two neuron types provided further evidence of functional cell-type-specific effects of tFUS.

## Discussion

In the present study, we set out to use multi-channel intracranial recordings to test our hypotheses of tFUS's ability to induce functional neuron-type-specific stimulation. Based on the results reported above, we have gained a better understanding of the tFUS parameter space, and thus may infer the mechanism of tFUS stimulation. Besides macroscopic perspectives reported in literature, uncovering the underlying neuronal mechanism requires a detailed inspection of how neurons respond to a vast set of acoustic parameters. Recordings using multi-channel intracranial electrophysiology allows us to examine the neuron cellular dynamics with high-spatial and temporal specificity.

**The selectivity between functional neuronal populations by tuning PRF.** Our results reported above provide in vivo evidence that subsets of neurons, functionally identified by their action potential waveforms (Figs. 3–6) and genotypes (Fig. 7), respond differently to the PRFs of tFUS stimulation. This provides insight into results previously reported by other groups studying tFUS[21,23,27,53,61]. The intrinsic different ion channel dynamics[52] and/or neuronal morphologies, such as profiles of dendritic arbors, of the different functional neuron types may be responsible to their unequal responses to the PRF change. This merits further investigations.

The ultimate goal of studying the mechanisms of tFUS is to translate the technology to clinical utility. Previous studies showed that inhibitory effect of tFUS was found at the primary somatosensory cortex[21] and thalamus[62] in humans, in which the same ultrasound parameters were employed (single-element transducer with $f_0 = 500$ kHz, PRF = 1 kHz, UD = 500 ms, DC = 36%). In contrast, other studies on humans showed excitatory effects that the primary visual cortex was directly excited with a specific ultrasound administration ($f_0 = 270$ kHz, PRF = 0.5 kHz, UD = 300 ms, DC = 50%)[27], and simultaneous sensory-evoking

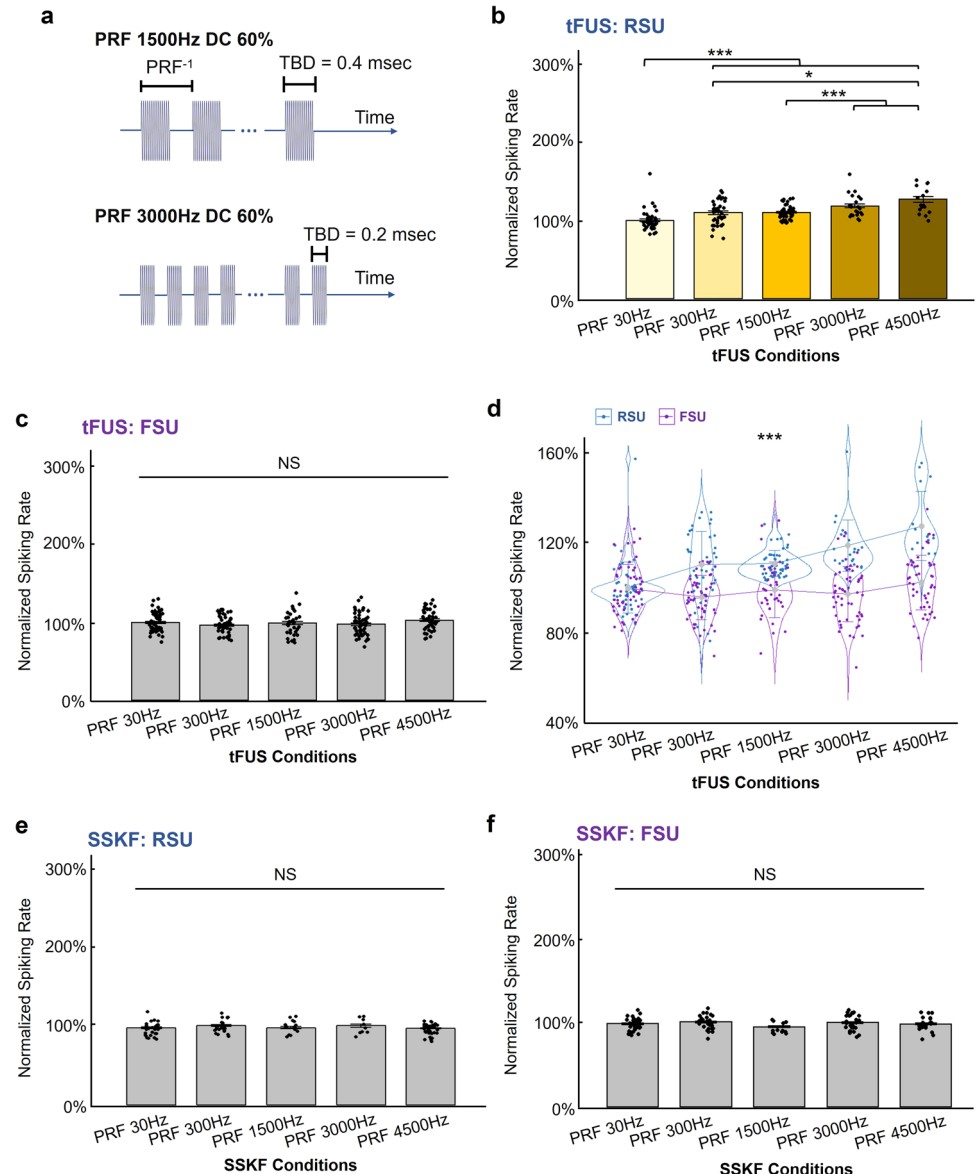

**Fig. 6 Validation of cell-type-selective responses to PRF in a constant duty cycle. a** The temporal profiles of the specific tFUS waveforms employed in this study. Stimulation parameters are listed in detail in Table 2. Two examples of PRF levels are shown. Number of cycles per pulse for PRF 1500 Hz and 3000 Hz are 200 and 100, respectively, with associated TBD of 0.4 and 0.2 ms. $f_0 = 500$ kHz, UD = 67 ms, ISoI = 2.5 s with 10% jittering. **b, c** The comparison for identified 417 single units across different PRF conditions under tFUS conditions with isoflurane anesthesia. 174 RSUs and 243 FSUs were separated. During tFUS condition, RSUs significantly differed their responses to different tFUS conditions (Kruskal–Wallis chi-squared = 70.61, $p = 1.69 \times 10^{-14}$). Data are shown as the mean ± s.e.m. of normalized firing rates, with statistical comparisons made through Kruskal–Wallis one-way ANOVA on ranks and post hoc one-tail two-sample Wilcoxon tests with Bonferroni correction for multiple comparisons (PRF 30 Hz vs. PRF 300 Hz: $p = 4.52 \times 10^{-5}$; PRF 30 Hz vs. PRF 1500 Hz: $p = 5.76 \times 10^{-11}$; PRF 30 Hz vs. PRF 3000 Hz: $p = 4.18 \times 10^{-9}$; PRF 30 Hz vs. PRF 4500 Hz: $p = 2.86 \times 10^{-8}$; PRF 300 Hz vs. PRF 4500 Hz: $p = 1.05 \times 10^{-3}$; PRF 1500 Hz vs. PRF 3000 Hz: $p = 9.30 \times 10^{-5}$; PRF 1500 Hz vs. PRF 4500 Hz: $p = 7.92 \times 10^{-7}$). *$p < 0.005$, ***$p < 0.0001$. NS, not significant. s.e.m., standard error of mean. **d** In a two-way ANOVA analysis, a statistically significant interaction ($p = 4.51 \times 10^{-10}$) between the neuronal type, i.e., RSU/FSU, and the PRF levels was found. Data are shown as the mean±s.d. of normalized firing rates. ***$p < 0.001$. s.d., standard deviation. **e, f** RSUs and FSUs showed homogeneous lack of response to the SSKF conditions. 120 RSUs, 128 FSUs. No significant effect of the ultrasound conditions was observed through statistics by Kruskal–Wallis two-sided *H*-test. Data are shown as the mean ± s.e.m. NS, not significant. s.e.m., standard error of mean. Source data are provided in the Source Data file.

capability of tFUS were also shown at primary and secondary somatosensory cortices ($f_0 = 210$ kHz, UD = 500 ms)[63]. Given these parameter-dependent studies, the effects of tFUS achieved by these two groups were observed to be inconsistent or even contradictory on healthy, awake subjects. We cannot conclude from these studies whether the observed behavior was due to overall activation or suppression of neural activity due to changes in tFUS parameters, or if the ultimate behavior was due to certain

selective modulation of the neural network. A modified theoretical model[25] was proposed in order to unify the ultrasound parametric space and predict either excitatory or inhibitory effects at a neuronal level. As we set out to explore the tFUS parameter space in the in vivo brains, we controlled PRFs and thus the duty cycles of tFUS at five levels (see Table 1 in Methods) while maintaining the $I_{SPPA}$, $I_{SPTP}$, and the tone-burst duration as constants. When tuning tFUS parameters, we discovered that

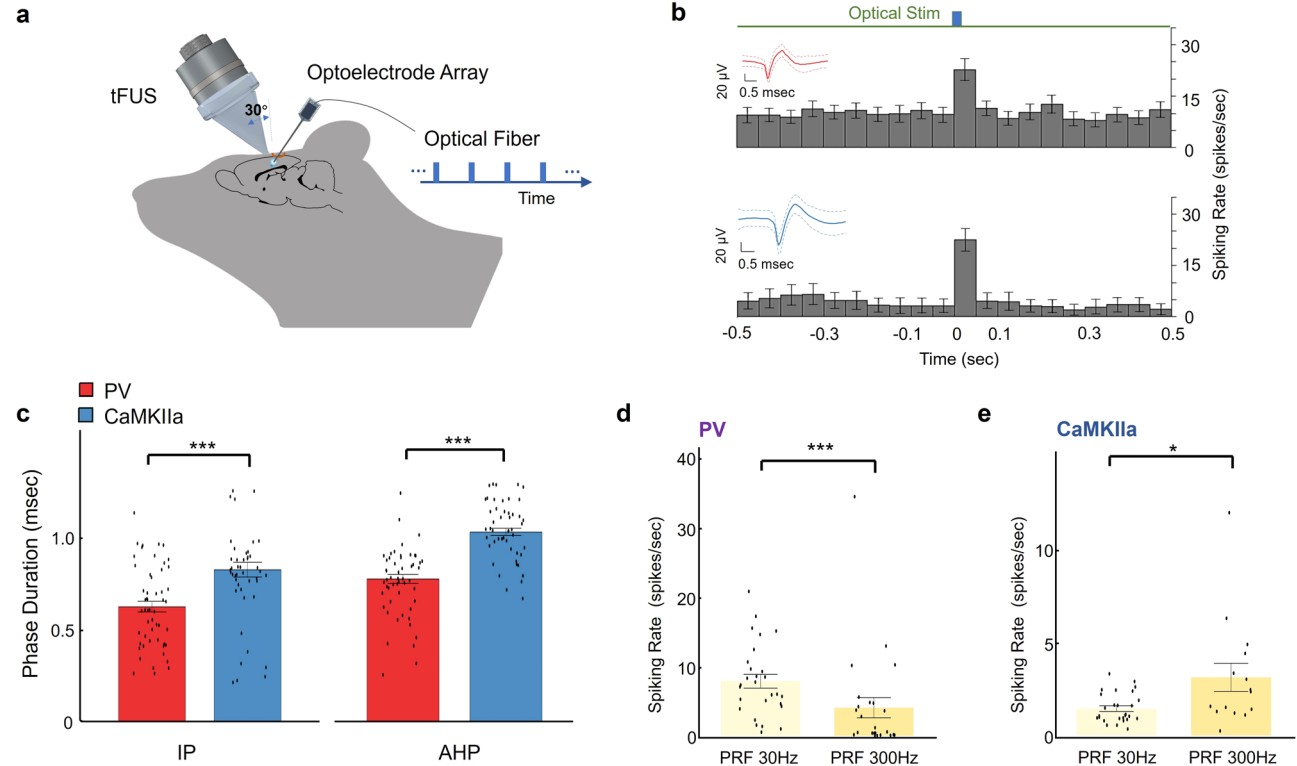

**Fig. 7 Validation of PRF preferences of inhibitory and excitatory neurons. a** A mouse-specific 3D-printed collimator guiding tFUS to hair-removed scalp of an anesthetized transgenic mouse model (2% isoflurane, 2 mg/kg bupivacaine subcutaneously) with an incidence angle of 30°, and a 32-channel optoelectrode array inserted at another incidence angle of 40° into the left S1 prepared through a craniotomy. The blue light stimuli were pulsed using a PlexBright LD-1 Single-Channel LED driver at 30 mA and were then delivered via optical fiber (105 μm diameter) to the recoding side of the shank. **b** The PSTHs (bin size: 50 ms) of one PV spiking unit (upper panel, 258 trials for each time bin, IP duration mean: 550 μs; AHP duration mean: 850 μs) and one CaMKIIa spiking unit (lower panel, 128 trials for each time bin, IP duration mean: 600 μs; AHP duration mean: 1100 μs) responding to the optical stimuli. Spike waveforms depicted as insets in **b**, solid red line as the mean waveform, dashed lines as the waveform standard deviation. Data are shown in the PSTHs as the mean ± 95% confidence interval. **c** Comparing PV ($n = 53$) and CaMKIIa ($n = 50$) neurons regarding the IP (PV vs. CaMKIIa: $p = 6 \times 10^{-5}$) and AHP (PV vs. CaMKIIa: $p = 3 \times 10^{-12}$) phase durations. Data are shown as the mean±s.e.m., statistics by one-tail Wilcoxon test. ***$p < 0.001$. **d, e** The comparisons of spiking rates between the tFUS conditions of PRF 30 Hz ($n = 28$ for PV neurons, $n = 27$ for CaMKIIa neurons) and PRF 300 Hz ($n = 25$ for PV neurons, $n = 15$ for CaMKIIa neurons) on PV (**d**) (PRF 30 Hz vs. PRF 300 Hz: $p = 2 \times 10^{-4}$) and CaMKIIa (**e**) (PRF 30 Hz vs. PRF 300 Hz: $p = 0.016$) neurons, respectively. Data are shown as the mean±s.e.m., statistics by two-tail Wilcoxon test. *$p < 0.05$, ***$p < 0.001$. Source data are provided in the Source Data file.

neuronal units grouped based upon action potential shape characteristics displayed different spiking rate during the same tFUS stimulation. The RSUs (presumably excitatory neurons) exhibited higher spike rates when stimulated with high PRF, thus high duty cycle, whereas the FSUs (presumably inhibitory neurons) exhibited homogeneous, high spike rate during the ultrasound stimulation across all PRFs. The inhibitory phenomena observed by Legon et al.[21,62] resulted from a DC located in a transition zone between tFUS induction of inhibitory and excitatory effects[25], whereas the brain activation reported by Lee et al.[27,63] was likely due to the applied PRF-related higher DC. In other words, the DC of 50% significantly increased the activity of excitatory neurons, and since the spiking activities of the inhibitory neurons did not increase proportionally in response to tFUS, therefore the overall behavioral outcomes resulted in facilitation. In order to decouple the PRF from the DC, we further investigated such neuron-type-specific responses to the changing PRF while maintaining the DC as a constant (i.e., 60%, see details in Table 2). The statistical results (Fig. 6b–d) validated the critical role of the PRF in tuning the neuronal responses. One should aware that the ultrasound tone-burst duration were changed accordingly in these experiments to match the same duty cycle.

**PRF: possible mechanism of the functional neuron-type-specific effects.** The PRF, one type of modulation on the ultrasound temporal wave, may lead to dynamic acoustic radiation force (ARF)[64]. In recent studies, the ARF were inferred as the most probable energy form that induced PRF-dependent behavioral responses[26] and the calcium signaling[61]. In the present study, when different PRFs were used to stimulate cortical neurons, we observed significant different responses of two neuronal subpopulations. We also considered that the difference in response between different neuron types was observed due to the interactions between transcranial ARF and ion channels in the neuron membrane. Neurons exhibit different action potential waveforms due to the difference in distribution of membrane proteins both in channel types and relative quantity of each type of ion channel. These distinct types of membrane proteins may have different response dynamics to acoustic radiation force[59,65]. We numerically calculated the ARF within a tFUS spatial-peak volume (e.g., 27 mm³), and the ARF ranged from 10.9 nN to 1.65 μN corresponding to the five levels of PRF (Supplementary Table 1). This range was in line with the previous reported values[26,61]. The basis of our hypothesis was tested between the FSUs and RSUs in the rat S1 cortex, through two different anesthesia methods, ketamine/xylazine cocktail and isoflurane.

We hypothesized that different types of neurons have distinct response profiles to the dynamic acoustic radiation forces exerted by the PRF of tFUS. The differences observed between different cell types may be attributed to the types and relative distributions of ion channels in each neuron-type and/or the distinct shape and orientation of the axonal and dendritic arbors, and such a hypothesis is illustrated in Supplementary Fig. 7.

We further tested this hypothesis in the S1 cortex of transgenic mice. Optogenetics was used to identify excitatory neuron and inhibitory neuron populations, by coexpressing channelrhodopsin only in CaMKII-alpha or PV expressing neurons. Both excitatory (CaMKII-alpha) and inhibitory (PV) neurons in response to the 30 and 300 Hz PRFs were significantly contrasted (Fig. 7d, e) comparing to the observations from non-transgenic rats (Fig. 4b, c). As a result, the fast-spiking units, presumably as inhibitory neurons, did not demonstrate a significant decrease of spiking rates when the PRF increased from 30 to 300 Hz ($p >$ 0.05). Although we cannot verify whether FSUs and RSUs corresponded directly to PV and CaMKII-alpha neurons in this study, this model still allowed us to study how specific protein-featured neurons respond to tFUS stimulation. Overall, in this optogenetic model, we also observed a distinctly unequal spiking response to tFUS stimulation PRFs. Thus, in two in vivo models and using two different methods to identify functional neuron types, we observed distinct responses to the PRF change of tFUS stimulation. Based on our findings, future investigations should use genetic approaches to attribute the observed functional cell-type-specific response to differences in protein expressions in different neuronal types.

**Controls for confounding effects**. Since our stimulation location was at the primary somatosensory cortex, control studies were conducted to examine whether activations recorded in the S1 could be due to somatosensation rather than direct activation of neurons in the S1. In a study examining the effect of tFUS on anesthesia recovery time, described in supplementary materials (Supplementary Fig. 8, Supplementary Note 1), rats with tFUS stimulation directly at S1 (PRF = 1500 Hz) recovered from anesthesia significantly faster than rats during sham conditions. Controls were tested on rats for auditory percepts coupled and not coupled to the skull, and rats with peripheral electrical stimulation at the contralateral hindlimb to control for somatosensation. Another extensive study was conducted to examine whether the induced neural activations is locally induced by tFUS, not due to confounding auditory effects (Supplementary Figs. 9, 10 and Supplementary Note 2). In both chemically deafened and genetically deafened rodents, local activations were observed through the LFP when tFUS was directed at S1. Collectively, these studies suggested that our tFUS experimental setup can elicit direct responses at the rodent brains without dependence on somatosensation and auditory percepts.

Another confounding factor we carefully controlled for was the interaction between the intracranial MEA and the tFUS field. Intracranial electrode arrays were widely used in order to record brain activity in high fidelity[66,67]. Many groups[29,43,50,53,68,69] employed intracranial recording probes in the examination of brain responses to focused ultrasound neuromodulation. The dimensions of these MEA electrodes are on the order of hundreds of microns, and typically on a rigid silicon or metal substrate. Although these MEAs are typically much smaller than the ultrasound wavelength, in the millimeter scale, there are possible locally enhanced ultrasound fields at the tissue interface of brain and electrode due to the large difference in ultrasound reflection coefficient. Furthermore, the interaction of the MEA electrode with the ultrasound pressure waves may produce indirect mechanical stimuli that leads to local neuronal activations. To control the mechanical vibration in MEA shank, we implemented multiphysics modeling and simulations to estimate the physical displacement of the MEA (Supplementary Fig. 11, Supplementary Note 3, Methods and Supplementary Materials for more details) and performed control experiments to compare the effect of low-intensity tFUS directed to the recording sites on the MEA or at the rigid electrode shank away from the brain and recording neuron population shown in Fig. 5 and Supplementary Fig. 6. As seen in Supplementary Fig. 6a–d, the increased neuronal firing aligned to tFUS stimulation was only observed when tFUS is directed at the recording neuron population, and the mechanical vibrations on the electrode shank (Fig. 5e–h and Supplementary Fig. 6e–h) did not lead to the similar neuronal effect as in the tFUS condition.

In this study, we conducted thorough sham studies to control for potential false-positive findings due to skull vibrations or vibrations of the recording electrode. Although we did not observe significant activations at auditory cortex (Supplementary Fig. 2a–c) when the low-intensity tFUS was targeted at S1, the potential auditory confounding factor still needs to be carefully controlled. The Supplementary Note 2 "Local Activations are Preserved in tFUS When Auditory Pathway is Blocked" included experimental details and results from chemically deafened rats and genetically deafened mice. Here, we provided one of the possible explanations to the previously observed auditory confounding effects[60,69], where observations in small rodent models might be confounded by extensive ultrasound standing wave field within the rodent skulls. We speculated that auditory activations may be due to ultrasonic standing waves in the skull cavity resultant from reflections at the skull base. Such wave pattern appeared to be more extensive when using even lower $f_0$[69] and more significant when the cranial bone was removed[69] (Supplementary Fig. 1c, d). Conventional ex vivo pressure mapping did not easily capture these standing waves due to the negligence of the major reflections at the skull base. Simulation results illustrated the estimated distribution of these standing waves (Supplementary Fig. 1), including the open-skull simulations (Supplementary Fig. 1c, d) using the equivalent head size of Wistar rat, given that the guinea pig models[69] have a similar skull size[70] as Wistar rats. In the small skull cavities, the ultrasound pressures were concentrated at the locally targeted region as expected using the angled incidence (Fig. 2), however, lower amplitude standing waves were still observed to be an extensive existence throughout the brain, including the auditory cortex especially when lower fundamental frequencies of 220[69] and 350 kHz[71] were employed as depicted in Supplementary Fig. 1a–d and e, f, respectively.

**Ultrasound safety**. All tFUS stimulation parameters used on the S1 cortices of rats and mice were maintained in brief exposures (i.e., 67-ms sonication per trial, with the total duty cycle of each trial being <3%) and with low intensities, which led to negligible calculated temperature rises (< 0.1 °C) at the targeted brain area. In addition, the mechanical index (MI) used in these experiments was less than 0.15, given the low peak negative pressure (i.e., <100 kPa). Such low MI made cavitation in brain tissue unlikely. These levels were well within the levels advised by the Food and Drug Administration (FDA) standard for ultrasound diagnostic imaging safety[72,73]. Moreover, hematoxylin and eosin stains gathered immediately after stimulation at S1 showed no evidence of neuronal damage, local hemorrhage or inflammatory response at the stimulation site (Supplementary Fig. 12). Additional in vitro examination is included in the note of Supplementary Fig. 12.

**Study limitation and future investigations**. In this work, the rodent models were sedated by anesthetic agents, which may introduce an inevitable confounding factor of changing the neuronal spiking activities. In particular, the injection of ketamine/xylazine did not provide a constant anesthesia level, which was why we did not observe a significant interaction between cell type and PRF levels initially, and thus why we also repeated our stimulation under isoflurane anesthesia and normalized to the trail-specific baseline. We randomized the order of applied tFUS and sham ultrasound conditions, which was believed to mitigate the influence of anesthetic level to the statistical analyses. Note that with the ketamine/xylazine anesthesia, it is possible that the FSUs' lack of significant response to the ultrasound PRF change was due to the limited sample size (Fig. 4c). To address this sampling size issue, we changed the anesthesia method to isoflurane in the follow-up studies (Figs. 5 and 6). In these studies, we were able to record from more FSUs. Similar observations were acquired regarding the lack of significant responses of FSUs to different levels of ultrasound PRF. Given the consistent observations in Figs. 4b, c, 5 and 6, we believe that the different responses observed from RSUs vs. FSUs to various PRF levels are not due to the sample numbers.

Caution should be taken when comparing between mice and rat models under tFUS stimulation. Our major constraint was due to the lack of widely available, well studied transgenic rat models for optogenetic stimulation. The thickness of the mice skull over the S1 cortex was several times thinner than that of rats. This leads to different acoustic insertion loss of tFUS field, which may result in differences during stimulation. For mice subjects, a different collimator with a smaller tip size was used (see Methods) to account for a smaller S1 region and avoid stimulating a widespread area in the mouse cortical brain. Discrepancies in activation area may also contribute to confounding results.

Regarding the effective transmission of ultrasound energy, besides restricting the size of collimator outlet to be commensurate with the ultrasound wavelength, we were also cautious about the potential effect of using the 2-mm burr hole (Fig. 1d) via which the electrode array was able to reach the neurons. For this reason, we introduced the 3D ultrasound field mappings (Fig. 2a, b) experimentally to explore potential alterations due to the low-acoustic-impedance conduit. Moreover, due to the requirement of 3D scanning of ultrasound field, we were only able to place the needle hydrophone (50 mm length) behind a freshly excised cranial piece in water, rather than inside a complete rodent skull and conducted the volumetric pressure mapping allowing ultrasound to reflect from the skull base. However, the latter one was believed to be more demanded so as to obtain additional knowledge of how standing waves appear inside the rodent skull cavity by using the 500 kHz fundamental frequency, given that considerable interference patterns due to standing waves were reported by administering 320 kHz tFUS to rats[74]. The 3D computer simulations (Fig. 2d, e) were included to further depict and investigate the possible wave patterns inside the small skull structure while adapting the applied ultrasound parameters. To our knowledge, this is the first simulation of tFUS field that takes the ultrasound guidance with the acoustic collimator into account.

A potential confounding factor to the above experiments was the lack of control for the effects of synaptic transmission between local RSU and FSU neurons of interest and the upstream innervation from surrounding areas. Although unlikely to be present in the current data due to averaging of large number of test trials (e.g., >400) and short spike response time, the possible effect of cross-cortical or inter-cortical communication cannot be exclusively discounted from the current experiment setup. Moreover, further fine tuning of tFUS stimulation parameters

may be needed to achieve increased differential activation for RSUs and FSUs. Given the validation results from optogenetic mice models (Fig. 7d, e), the functional separation between RSUs and FSUs is deemed as a broad but initial finding of the intrinsic functional cell-type selectivity of tFUS.

A future study would be beneficial by substantially characterizing how the continuous wave (CW) ultrasound configuration would impact on the intrinsic neuron-type selectivity of the tFUS, while the temperature rise due to the CW needs to be carefully controlled and minimized in order to avoid local temperature changes, which are known modulators of neural excitability[75,76]. Furthermore, as reported previously[53,60], the angles of the ultrasound incidence were designed to physically accommodate the ultrasound apparatus and the recording probe; however, we tried to preserve the ultrasound longitudinal wave rather than those in the shear mode. Although it is unavoidable in the practice, the angled tFUS may introduce nontrivial shear wave propagating along the skull, thus leading to increased skull conduction. Nevertheless, the difference in the ultrasound wave mode might result in differed neuronal responses, which also necessitates further investigations.

## Methods

### Experimental model and subject details

*Rat subjects*. Wistar outbred male rats (Hsd:WI, Envigo, USA) were used as subjects, and all rat studies were approved by the Institutional Animal Care and Use Committee at the University of Minnesota and Carnegie Mellon University in accordance with US National Institutes of Health guidelines.

*Transgenic mouse subjects*. Transgenic mice models were purchased from The Jackson Laboratory and bred to achieve the desired strains. CaMKIIa-ChR2 mice were crossed between a T29-1 parent expressing calcium/calmodulin-dependent protein kinase II alpha (CaMKII-alpha) promoter driving Cre recombinase expression, and an *Ai32* parent, expressing channelrhodopsin-2/EYFP fusion protein following exposure to Cre recombinase. A genetically positive offspring coexpressing ChR2 in CaMKII-alpha neurons were identified via tail snip DNA testing using YFP as a probe. PV-ChR2 mice were crossed between a PV-Cre parent expressing Cre recombinase in parvalbumin-expressing neurons and an *Ai32* parent. A genetically positive offspring coexpressing ChR2 in PV neurons were identified via fluorescent protein visualization goggles (BLS, Budapest, Hungary). The mice were housed at 22–23 °C (relative humidity: 30–70%) on a 12-h light/dark cycle. The procedures were reviewed and approved by the Institutional Animal Care and Use Committee at the University of Minnesota in accordance with US National Institutes of Health guidelines.

*Genetically deafened mice subjects*. Mice with homozygous expressions of the naturally occurring *Atp2b2* or *Pmca2* gene mutation have a deaf phenotype accompanies by unstable gait, commonly referred to as "the deaf waddler". The *Atp2b2* or *Pmca2* gene encodes for plasma membrane calcium-transporting ATPase 2 protein. Mice subjects were purchased through cryo recovery (Stock No. 001276, The Jackson Laboratories, USA). Mice were bred and housed at 22–23 °C (relative humidity: 30–70%) on a 12-h light/dark cycle at Carnegie Mellon University. All mice studies were approved by the Institutional Animal Care and Use Committee at Carnegie Mellon University.

### Method details

*tFUS setup and parameter selection*. Single-element focused transducers were used for tFUS stimulation with specifications of acoustic aperture outer diameter (OD) 28.5 mm, ultrasound fundamental frequency ($f_0$) 0.5 MHz, −6 dB bandwidth 300–690 kHz, a nominal focal distance of 38 mm (V391-SU-F1.5IN-PTF, Olympus Scientific Solutions Americas, Inc., USA) or transducer OD 25.4 mm, $f_0$ 0.5 MHz, a nominal focal distance of 38 mm (AT31529, Blatek Industries, Inc., USA). Collimators were 3D-printed with VeroClear material to match the focal length of the transducer and the animal model, the outlet of the angled collimator for the rat model has an elliptical area of 25.6 mm² (major axis length: 6.8 mm, minor axis length: 5 mm), and the one for the ultrasound normal incidence has a circular area of 19.64 mm². The size of collimators' outlet was set to be no less than or at least commensurate with one ultrasound wavelength (i.e., 3 mm in soft tissue). One single-channel waveform generator (33220 A, Keysight Technologies, Inc., USA) was working with another double-channel generator (33612A, Keysight Technologies, Inc., USA) to control the timing of each sonication, synchronize the ultrasound transmission with neural recording, and form the initial ultrasound waveform to be amplified, thus driving the transducer. A 50-watt wide-band radio-frequency (RF) power amplifier (BBS0D3FHM, Empower RF Systems, Inc., USA) was employed to amplify the low-voltage ultrasound waveform signal. The

**Table 3 Acoustic parameters for numerical simulations.**

| Materials | $c$ (m/s) | $\rho$ (kg/m$^3$) | $\alpha_0$ (dB/MHz/cm) |
|---|---|---|---|
| Water | 1482 | 1000 | $3.02 \times 10^{-5}$ |
| Cortical bone | 3100 | 2200 | 1.87–18.15 |
| Collimator[a] | 2410 | 1160 | 9.54 |

[a]Collimator 3D-printed using VeroClear$^{TM}$. $c$: speed of sound, $\rho$: mass density, $\alpha_0$: ultrasound attenuation coefficient.

employed ultrasound intensity levels and duty cycles were described in Table 1. As noted in Table 1, all ultrasound conditions used the same $f_0$ of 0.5 MHz, ultrasound duration (UD, i.e., sonication duration) of 67 ms and tone-burst duration (TBD) of 200 μs. In Table 2, the ultrasound parameters for those five conditions were to keep the ultrasound duty cycle (DC, within the ultrasound duration of 67 ms) constant at 60%. This set of parameters also maintained equivalent spatial-peak temporal-average intensity ($I_{SPTA}$), respectively, among PRF levels.

*Extracellular recordings.* Extracellular recordings were implemented using 32-channel 10-mm single shank electrodes, where electrode sites were arranged in three columns, spaced 50 microns apart from each other (A1x32-Poly3-10mm-50-177, NeuroNexus, Ann Arbor, MI, USA). Electrodes were inserted into the rodent skull using a small animal stereotaxic frame with 10-micron precision manipulators (Model 963, David Kopf Instruments, Tujunga, CA, USA). Electrodes were inserted at a 40-degree angle in the sagittal plane.

Rodents were sedated using either ketamine/xylazine cocktail for results presented in Figs. 3b–g and 4 or isoflurane for results in Figs. 3a, 5, 6, and 7. All animals were skin prepared with hair removal gel. Rodent heart rate, respiration rates and rectal temperatures were monitored throughout recordings. Cranial windows, 1–2 mm in diameter, were opened in the skull using a high-speed micro drill (Model 1474, David Kopf Instruments, Tujunga, CA, USA) under stereotaxic surgery assisted with microscope system (V-series otology microscope, JEDMED, St. Louis, MO, USA). Skull suture lines were used to identify brain structure locations. Recordings were obtained using the NeuroNexus Smartbox recording system (20 kHz sampling frequency, NeuroNexus, Ann Arbor, MI, USA) and the TDT Neurophysiology Workstation (24 kHz sampling frequency, Tucker-Davis Technologies, Alachua, FL, USA).

*Stimulation at primary somatosensory cortex in rats.* Rat subjects were adults in the weight range of 350 to 550 g, initially sedated under ketamine/xylazine cocktail (75 mg/kg, 10 mg/kg) and extended with ketamine injections (75 mg/kg). This sedative approach allowed us to achieve stable anesthesia with minimum body movement, e.g., breathing, heart beating, etc. For validation purposes (Figs. 5, 6, and S5), the rat subjects were anesthetized using 4% isoflurane and O$_2$ mixture for induction and were kept sedated using 1.5-2% isoflurane during recordings. The vital signals, e.g., breathing and heart beating rates, body temperature, were monitored and recorded during the stimulations. tFUS collimator was coupled with the rat skull using ultrasound transmission gel (Aquasonic ultrasound gel, Parker Laboratories, Inc., Fairfield, NJ, USA). Ultrasound wave were transmitted through intact skull, directed at 40-degree angle towards the left S1 region on the rat head.

*Stimulation at primary somatosensory cortex in optogenetic mice.* Mice expressing ChR2(H134R) in select neuronal populations were achieved by crossing mice expressing ChR2 in a cre-dependent manner (B6;129S-Gt(ROSA)26Sortm32.1 (CAG-COP4*H134R/EYFP)Hze/J; Jackson labs stock number: 012569; also referred to as Ai32; donated to Jackson labs by Hongkui Zeng) with mice expressing Cre recombinase in either parvalbumin-expressing cells, including fast-spiking inhibitory interneurons (B6;129P2-Pvalbtm1(cre)Arbr/J; Jackson labs stock number: 008069[77]; donated to Jackson labs by Silvia Arber; resulting cross referred to as PV-ChR2) or in neurons expressing CaMKIIα, including RS excitatory cortical neurons (B6.Cg-Tg(Camk2a-cre)T29-1Stl/J; Jackson labs stock number: 005359[78]; donated to Jackson labs by Susumu Tonegawa; resulting cross referred to as CaMKII-ChR2). Opsin-expressing offspring were identified by genotyping (for CaMKII-ChR2 mice; Transnetyx) or via fluorescent protein visualization goggles (BLS, Budapest, Hungary) (for PV-ChR2 animals), as done previously[79].

Recordings were performed in adult mice, males and females, older than 8 weeks old. Mice were sedated under isoflurane (4% during surgery, 2% during recording) and 2 mg/kg bupivacaine subcutaneously. Optoelectrodes (NeuroNexus, Ann Arbor, MI, USA) were inserted in the S1 (ML: −1.5 mm, AP: −1 mm, Depth: 0.7 mm) at a 40-degree angle from the posterior, while tFUS was delivered at 30-degree incidence angle from the anterior in the same sagittal plane. After all tFUS stimulations were performed, to avoid confounds of light delivery potentially altering neuronal firing properties[80,81], brief pulses of light (wavelength 465 nm, ≤10 ms duration, the duration of light was adjusted to avoid bursts of activity; PlexBright LED Module, Plexon, Dallas, TX, USA) were used to "optotag" recorded neurons. Units which displayed an increase in firing rate (defined as an increase above 99% confidence interval of spontaneous firing) within 10 ms of the light

pulse were classified as excitatory neurons (in CaMKII-ChR2 mice) or parvalbumin-expressing inhibitory neurons (in PV-ChR2 mice).

*Histology.* Explanted rodent brains were fixed in 5% formaldehyde for at least 48 h and stored in 70% ethanol until samples were ready for histology. Samples were then dehydrated and embedded in paraffin for sectioned coronally at 5–10 micron per section. Tissue samples were then stained with Haemotoxylin and Eosin (H&E) by the Histology & Research Laboratory at the University of Minnesota.

**Quantification analysis**

*MUA and LFP data processing.* For spike analysis, neural traces were band-pass filtered between 244 Hz and 6 kHz. In some of the noisy datasets, the band-pass filtered MUA data were further processed in order to increase the signal-to-noise ratio (SNR). The potential recording artifacts existing in the MUA were identified by computing the inter-spike interval (ISpI) and examining its temporal statistical distribution presented in the return plots (i.e., Poincaré plot, a second-order analysis method for nonlinear features in time series), e.g., Supplementary Fig. 4b, d, f and 5b, c. In the cases of noisy datasets, Symlet wavelet denoising was applied to all the recording channels using Wavelet toolbox in MATLAB v9.0.0 (The MathWorks, Inc., Natick, MA, USA) to remove the potential artifacts, such as electromagnetic interferences, inefficient ground coupling, and possible mouth movement (e.g., licking). All MUA spike sorting and single-unit preselection were performed using PCA-based spike classification software Offline Sorter (Plexon, Dallas, TX, USA). Local field potentials (LFP) were band-passed from 1 Hz to 244 Hz. The band-pass filtered LFP data were then averaged across trials time-locked to the ultrasound onset with trial numbers indicated with the descriptions of respective sub-studies. In the case of obvious pre-stimulus fluctuations, the LFP data were further denoised using Wiener filter and independent component analysis in the MATLAB to generate Supplementary Fig. 10c. Further analyses, including the ISpI computation, PSTH, raster and return plots, feature extraction for the initial phase (IP) and afterhyperpolarization (AHP), descriptive statistics for spike waveform and spike rates, LFP temporal and spectral analyses were performed using FieldTrip toolbox[82] or custom code in the MATLAB. After obtaining phase durations of IP and AHP, $k$-means clustering was employed to conduct the cluster analysis of neuron types in the MATLAB. Further statistical analyses were performed in R (V3.2.1 and V3.5.3, The R Foundation, https://www.R-project.org/).

*Ultrasound pressure/intensity mapping.* In order to characterize tFUS stimulation's temporal and spatial dynamics near the brain targets, we developed a three-dimensional ex vivo pressure mapping system that used a water submerged needle hydrophone (HNR500, Onda Corporation, Sunnyvale, CA USA) driven by a 3-axial positioning stage (XSlide, Velmex, Inc., Bloomfield, NY, USA) to map out the spatial-temporal pressure profiles of ultrasound transmitted through an ex vivo skull. The needle hydrophone was placed beneath an ex vivo skull and recorded ultrasound pressure values at discrete locations (scanning resolution: 0.25 mm laterally, 0.5 mm axially) behind the skull. Skulls were freshly dissected from euthanized animals. This setup allowed us to quantify the amount of energy delivered to the brain, which varied substantially due to the inhomogeneity and aperture of the skull. The system can be set up to mimic the exact conditions of the ultrasound set up with matched collimator locations and angles.

Based on the measured 3-D ultrasonic pressure map, the spatial-peak temporal-peak intensity ($I_{SPTP}$) was calculated using Eq. 1.

$$I_{SPTP} = \frac{P_0^2}{Z_0} \qquad (1)$$

where $P_0$ is the maximal instantaneous pressure amplitude in both spatial and temporal domains, and $Z_0$ is the characteristic specific acoustic impedance. This impedance can be computed using Eq. 2.

$$Z_0 = \rho \cdot c \qquad (2)$$

where $\rho$ is the medium density (1028 kg/m$^3$ for brain tissue, assuming 2200 kg/m$^3$ for cortical bone[83]), and $c$ is the speed of sound in the medium (1515 m/s for brain tissue, assuming 3100 m/s for cortical bone[84]).

We can also obtain the spatial-peak temporal-average intensity ($I_{SPTA}$) from temporal profile of ultrasound pressure at its spatial maximum, which is denoted as $P(t)$. Equation 3 was used to calculate the $I_{SPTA}$[73].

$$I_{SPTA} = \int \frac{P(t)^2}{Z_0} dt \cdot PRF \qquad (3)$$

The time window for the above integration is the pulse duration[29,53,73]. Moreover, the 3D computer simulations to study the potential ultrasound wave patterns residing in the small skull cavity were conducted using a MATLAB-based toolbox, k-Wave[85]. The computed tomography (CT) images of the Wistar rat were acquired from the scan using a Siemens Inveon micro-CT machine at the University Imaging Center of the University of Minnesota. The CT images of an adult mouse were acquired from Rangos Research Center Animal Imaging Core Facility in University of Pittsburgh. The simulation study was following a similar protocol described by Mueller et al.[86] in order to obtain the spatial distributions of medium density, speed of sound and acoustic attenuation throughout the rodent skull based on the porosity calculated from CT Hounsfield units[87] using the

assumed acoustic parameters listed in Table 3. The skull was deemed to be immersed in water and its acoustic properties were mapped from the Hounsfield unit acquired from the CT images and thus the porosity readings. In the simulations, the Courant-Friedrichs-Lewy (CFL) number, the ratio of the wave traveling distance per time step to the space grid, was set at 0.02 to ensure stability of the simulations.

*Ultrasound-induced temperature rise.* As a safety concern, once obtaining the ultrasound pressure map, we did a numerical estimation of maximum temperature rise using the following calculation methods[88]. First, we obtained the $I_{SPTA}$ using Eq. 3, and then the rate of heat generation per volume was calculated using Eq. 4.

$$\dot{Q} = 2\alpha \cdot I_{SPTA} \qquad (4)$$

where $\alpha$ is the ultrasound absorption coefficient in brain tissue (0.03 Np/cm at 0.5 MHz[89]) or in cortical bone (3.45 dB/cm at 0.5 MHz[90]). Since we introduced the spatial-peak temporal-average intensity, the estimation of the $Q$-value was maximized for the targeted site. Next, we borrowed the Eq. 5 from[91] to obtain the maximum temperature increase if we assume no heat removal process took place during the ultrasound energy deposition:

$$\triangle T_{max} = \frac{\dot{Q} \cdot \triangle t}{C_v} \qquad (5)$$

where $\triangle t$ is the tissue exposure time under tFUS, and $C_v$ is the heat capacity per unit volume (3.6 J/g/°C for the brain tissue, 1.606 J/g/°C for the skull[92]). We estimated the temperature rise at a disk-shape focal area with a radius of one ultrasound wavelength (i.e., 3 mm). With these, the estimation would produce an upper limit for the temperature change.

*Ultrasound-induced acoustic radiation force.* The magnitude of dynamic acoustic radiation force (ARF) generated by the PRF can be numerically estimated at the spatial peak of ultrasound field within the skull cavity using the simplified equation (Eq. 6)[93].

$$|\boldsymbol{F}| = \frac{2\alpha \cdot I_{SPTA}}{c} \qquad (6)$$

The 500 kHz acoustic absorption coefficient ($\alpha$) of brain tissue is 3 Np/m, the $I_{SPTA}$ is listed in Table 1 for all the five conditions at different PRF levels, and the speed of sound ($c$) in the brain tissue is 1515 m/s. The estimated ARF magnitude at a spatial-peak volume of e.g., 27 mm³ (i.e., 3 x 3 x 3 mm³, 3 mm is the approximate wavelength of 500 kHz ultrasound in the soft tissue) is listed in Supplementary Table 1.

*Quantify ultrasound-induced shank vibration on neuronexus electrodes.* A COM-SOL model was created to simulate electrode deflection properties while implanted in the brain and an ultrasound-induced ARF was being applied (Supplementary Fig. 11 and Supplementary Note 3). To model this scenario, we used the solid mechanics and fluid-structure interaction modules in COMSOL 5.5 (COMSOL, Inc., Burlington, MA). The details of this simulation were included in the Supplementary Note 3, "A Simulation of Electrode Vibrations During tFUS".

Based on the above computer simulations and the acoustic insertion loss due to the skull (−5.5 dB measured from ex vivo ultrasound pressure scans), 4 levels of ultrasound pressures were identified and applied to the electrode shank. 100% SSHK used the full ultrasound pressure as used in tFUS setup, while the other three SSHK levels administered the corresponding percentage, i.e., 43%, 28% and 13% of tFUS ultrasound energy onto the shank. We denoted those three sham conditions as 43% SSHK, 28% SSHK and 13% SSHK, respectively. In the SSHK conditions, as the setup shown in Figs. 1g and 5d, the ultrasound was delivered directly onto the middle of electrode shank without penetrating the skull. This was the first difference of SSHK from the tFUS setup shown in Fig. 1a. From Fig. 2b, c, the acoustic pressure insertion loss of the 500 kHz tFUS due to the skull bone was −5.5 dB. This was the first ratio that needs to be derated from the full ultrasound pressure transmitted in the tFUS setup. The second difference between these two setups was the locations on the electrode shank for sonication targeting, i.e., at shank tip in tFUS sonication as shown in Figs. 1a and 2d vs. at the middle of the shank in the SSHK condition (Fig. 5d). We thus numerically modeled the maximum electrode deflection displacement at the electrode tip using a cantilever model (https://www.engineeringtoolbox.com/cantilever-beams-d_1848.html) when the tFUS wave was targeting at the tip or at the middle part. This leads to the second ratio of 81%. We multiplied those two major ratios and obtained 43%. This number was a rough estimation to derate the full ultrasound pressure being transmitted in the tFUS setup for the sake of reproducing a roughly equivalent vibration at the electrode tip in both tFUS and SSHK conditions. To make this SSHK sham experiments a rigorous control study in order to eliminate the confounding factor of electrode vibration at the electrode sensing tip, we thus introduced those four pressure levels of SSHK, i.e., 100%, 43%, 28% and 13%, in order to cover the full range of possible electrode mechanical vibrations' amplitude due to the tFUS incidence.

**Reporting summary**. Further information on research design is available in the Nature Research Reporting Summary linked to this article.

## Data availability

The data that support the findings of this study are presented in the paper and supplementary materials. Source data are provided with this paper. Data of sorted neural spikes in 10 animals can be found from Figshare repository at: https://doi.org/10.6084/m9.figshare.14150336. Additional data are available from the corresponding author upon reasonable request. Source data are provided with this paper.

## Code availability

Codes for processing the normalized spike rate can be found from Figshare repository at: https://doi.org/10.6084/m9.figshare.14150336. Additional codes that support the findings of this study are available from the corresponding author upon reasonable request.

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

## Acknowledgements
This work was supported in part by NIH grants MH114233, EB029354, AT009263, EB021027, NS096761, and NSF grants CBET-1450956 and CBET-1264782. K.Y. was supported in part by The Samuel and Emma Winters Foundation, an MnDRIVE Neuromodulation Fellowship and Doctoral Dissertation Fellowship from the University of Minnesota. X.N. was supported in part by Liang Ji Dian Graduate Fellowship and Carnegie Mellon Neuroscience Institute Presidential Fellowship at Carnegie Mellon University. We thank Dr. Akira Sumiyoshi from Tohoku University for providing Wistar rat MRI atlas and Dr. Yijen Wu from Rangos Research Center Animal Imaging Core for providing micro-CT and MRI atlas of mice. We are also grateful to Drs. Abbas Sohrabpour and Haiteng Jiang for stimulating discussions, Dr. Qi Shao for coordinating histology studies, Dr. Yi Zhang for training in animal surgery, John Basile for assisting equipment setup, as well as Daniel Suma, Maryam Zhian, and Mckinney Zhang for assistance.

## Author contributions
Conceptualization, K.Y., X.N., and B.H.; Methodology, K.Y., X.N., and E.K.-M.; Formal Analysis, K.Y.; Investigation, K.Y., X.N., and B.H.; Resources, B.H., and E.K.-M.; Writing—Original Draft, K.Y. and X.N.; Writing—Review & Editing, K.Y., X.N., E.K.-M. and B.H.; Supervision, B.H.

## Competing interests
The authors declare the following competing interests: K.Y., X.N., and B.H. have a joint pending patent application with PCT (PCT/US2019/055955) on some techniques used in this investigation. E.K.-M. does not have competing interest.
