## [Peer Review File · Nature Communications]

Reviewer #1 (Remarks to the Author):

This is a lengthy article looking at ultrasound modulation of two populations of neurons with a multielectrode array. It is a tour-de-force study, with many aspects and definitely of reader interest. But, the length leads to a negative of the manuscript: it is hard to follow the rationale for experimental changes between the parts and hard to follow to follow the conclusions etc. I would think it would be better split up into 2 or more focused manuscripts. In addition, there are many parts that need clarification. I have highlighted many of them, but i fear that there are many more that also need clarification. I would encourage the authors to broaden the comments below into other figures/concepts that similarly need clarification.

It would be helpful to have scan tank pictures of the focal spot without the cranium, alongside Fig 2b. It's surprising to me that the rat skull would aberrate the beam enough to give such a weird spot shape in Fig 2b. Fig 2c and 2d are too dark to see the colors.

Were there any studies to demonstrate how much ultrasound penetrates when a 40° angle is use on the rat skull? It might be nice to have a pair of figures on this alongside those above.

The text says that intensities are provided in Table 1, but they are not.

When were ketamine injections done?

In the calculation for $I_{\text{sp}}t$, it's not clear why there is the factor of UPRF. It would seem that the integral over time would have sufficed. Maybe the confusion is because it does not specify the time of the integral.

Is the claim that the focus is within 1 mm of the skull because that is where the maroon color is? The focus is usually defined as the -3dB line, so that would be much bigger, in the green color.

Why was the sham SSHK not included in Fig 3A?

In Fig S2, it doesn't look like there is an increase in spiking in the live animal, like there is in Fig 1A. Why is that?

Fig 3 shows time locked signals in response to the ultrasound. But, it seems that the data in Fig 4 is not time locked data. Why? Why was time locked data shown in one Figure, then abandoned for the next figure?

Are there 146 dots in each of the bars of Fig 4B? Are there 53 dots in each of the bars in Fig 4C?

In Fig 4, what is the meaning of the horizontal variation within a UPRF category?

Why was the simulation in Fig S1 done at 220 kHz when the rest of the study was done at 500 kHz?

Where is Fig S2 E-F?

On page 11, it is not clear why the change in anesthesia was made. It may have changed everything and should have been kept constant.

In Fig 5, it is not clear what is meant by Ultrasound was delivered in two grades: 100% and 43%. 43% of what? Why 43%?

A significant limitation to this study is that the Ispta is not kept constant. This should be discussed appropriately in the discussion. As is, it does not really shed light on the effect of duty cycle seen by others, when they do keep Ispta constant.

Two anesthetics were used, but it is not clear why and what is the effect of doing so.

The manuscript is quite long and I wonder if it wouldn't be better broken up into separate manuscripts. The discussion is quite long and it is not clear that all the points are well supported. It is not clear that the mechanism discussion is supported by the data, at least this is not clearly laid out.

The question about the vibration is not clearly removed when the ultrasound is directed at the shank. It would be nice to have a better picture of the setup. More importantly, it would be nice to have an independent measure of the vibration of the MEA when the US field is directed at the tissue and when directed at the shank, to know if it is a good control.

Reviewer #2 (Remarks to the Author):

This is an interesting study that sought to understand the effect of ultrasound pulse repetition frequency (UPRF) on subtypes of cortical neuronal cells, namely excitatory or inhibitory. Two models were used (1 rat and 1 mouse) to functionally separate cells (Regular-spiking units RSU; excitatory vs Fast-spiking units FSU; inhibitory) or (PV interneurons; inhibitory vs CaMKII-alpha; excitatory). Lastly, an investigation of UPRF on lasting effects such as long-term potentiation (LTP), long-term depression (LTD), and short-term depression (STD) was conducted. The results presented in the study are well described and provide interesting connections to previous studies which may have wide reader interest. However, the study could benefit from clarification on the novelty and justification of the methodology as well as corrections to critical assertions. Specific comments are as follows:

1. Figure 3 - only two sham conditions were shown whereas 3 sham conditions were introduced in Figure 1. It would be interesting to see the MUA data for the shank sham condition, especially for the same anesthesia condition.
2. Figure 3 C,D,F,G bottom spike plots are hard to see
3. Introduction, first para: "designer receptors exclusively activated by designer drugs (DREADDs) etc., have been developed in order to modulate and study the brain" Please replace 'etc' with the actual other agents and provide corresponding references.
4. Page 4, 2nd para: Remove 'mechanical' after 'ultrasonic'
5. Page 5, 'Similar to electrical stimulation, we hypothesize electrical stimulation.' This hypothesis is not new and several others have already provided comparison of tFUS to electrical with several similarities. Please indicate the exact novelty of this study.
6. Page 7, last para: Indicate how artifacts were identified and/or removed at preprocessing.
7. Page 8, paragraph 1, line 4 – Ref 60-62 show RSU/FSU for activation of receptive fields rather than direct activation of the neurons themselves, since, in the present study, tFUS is acting on the neurons directly, how applicable is this metric?

8. Page 8, paragraph 1, line 9 – Ref 61 contradicts the authors' hypothesis that UPRF drives RSU/FSU through intrinsic cellular differences. Ref 61 states RSU/FSU have distinct responses not because of the intrinsic cellular differences but rather due to differences in the strength of the thalamocortical inputs targeting each cell type. Please address.
9. Page 10: 'The baseline firing rate of each single unit in.... tFUS conditions' Why not monitor anesthesia depth and relate it to the firing rate? Randomizing does not necessarily reduce the anesthesia effect as it also has a dynamic component.
10. Page 13, 1st para: Please provide evidence of standing waves. The 500 KHz frequency has a 3 mm wavelength and the mouse brain is typically 7 mm in depth, it is not clear how standing waves would be generated.
11. Page 13, 2nd para: How was the focusing performed in 'deep brain'?
12. Page 13: '180 degrees away from the skull' what does this mean?
13. Page 16, paragraph 1, line 4 – data showing that results suggest tFUS affects ion channel dynamics or neuronal morphology is missing.
14. Page 17, paragraph 1, line 2 – Ref 21 does not have to be placed twice within the same sentence
15. Page 18, paragraph 1 - Acoustic Radiation Force was hypothesized to be driven by UPRF, can you show measurements or models of the types of force your parameters produce?
16. Since the transgenic mice experiments are essential to the study, the figures should be important enough to move to the main text rather than the supplementary.
17. Page 19: replace 'sonication frequency' with 'pulse repetition frequency'
18. Page 20, paragraph 2, line 9 – NMDA receptors in LTD are still being activated. This is essential for moderate calcium influx so that downregulation of AMPA receptors can occur, decreasing sensitivity to glutamate.
19. It is unclear which area (deep brain) is stimulated.
20. Page 25, paragraph 1 – Why was live/dead analysis performed in vitro rather than in vivo?
21. Figure 7 – which sham condition is depicted?
22. Adjustment for isoflurane vs ketamine/xylazine may be important for other studies, please elaborate.

Reviewer #3 (Remarks to the Author):

This study performed several electrophysiological experiments in rats and mice to determine how specific neurons respond to specific ultrasound pulse repetition frequencies (PRF) and duty cycles (DC). The study is positioned timely within a broader quest of several groups to determine the most effective ultrasonic stimulation protocols and the associated mechanisms of action. The study found that the firing rates of specific neuronal pools can be influenced by PRF or DC.

Both PRF and DC are thought to be important factors in ultrasonic neuromodulation. By design, PRF and DC were confounded in this study (Table 1). Therefore, it is not possible to distinguish whether the observed variability in neuronal responses is due to PRF or DC. Consequently, it is difficult if not impossible to devise a model that can explain the effects.

This positions the study to be more descriptive rather than generating normative knowledge for the field.

However, the finding that distinct neuronal cell types respond to certain ultrasound parameters distinctly is still important for the field.

The findings of Fig. S9 D and E are particularly striking, showing a double dissociation of the effects of PRF (or DC) on putative excitatory and inhibitory cells. It is unfortunate that this approach was affected by noise, and so only low PRFs/DCs could be tested. In either case, I suggest to bring this striking finding early on in a future version of the paper, rather than leaving it for the end of the supplement.

The main claim about the influence of PRF/DC on neuronal firing is based on data shown in Fig. 4B, C.

This claim should be supported by a proper, omnibus test rather than the current multiple comparisons.

In particular, this should be tested using a 2-way ANOVA with factors [cell type], [PRF], and their interaction, [cell type]*[PRF]. If there is a statistically significant leverage of PRF/DC on distinct cells type, this would show as the significant [cell type]*[PRF] interaction term in this ANOVA model. Visually, this interaction does not appear significant in Fig. 4B, C. It does, however, appear significant under the different mode of anesthesia--isoflurane (Fig. 5B, C). To incorporate this, the factor of anesthesia could be included as a third factor in the ANOVA. One would still test for the interaction of [cell type]*[PRF] in this extended model.

Since the LTD effects (Figs. 6 and 7) do not show significant variability by PRF/DC, these data do not contribute to the present framing of the paper and so should likely be presented in a separate paper.

Minor suggestions

"UPRFxX" is cryptic. Why not show the specific frequency/duty cycle values, to spare the reader of having to refer to Table 1.

"ultrasound is delivered to the shank of the electrode upstream" - upstream is generally used in the context of the direction of flow of neural information; something like "at the electrode base" would be more descriptive.

typos:

"So far, none have explored"

"These are early evidences"

Response to Reviewers

We are extremely grateful to the editors and reviewers for their constructive comments. We have taken very significant efforts to conduct a series of additional experiments and analyses and revised the manuscript substantially to address the reviewers' comments. In particular, we have included new sham conditions experiments investigating the potential effects of electrode vibrations on neural activations, and presence of sustained neural effects in the hippocampus under a different anesthesia method using isoflurane gas. We believe the work has been significantly enhanced and improved.

Our responses to the reviewer comments are shown below in BLUE and the reviewer comments in BLACK. Fig. x refers the figures in the main manuscript, and Fig. Sx refers the figures in the supplements. Substantially revised or added texts in the manuscript are printed in RED.

Reviewer #1:

This is a lengthy article looking at ultrasound modulation of two populations of neurons with a multielectrode array. It is a tour-de-force study, with many aspects and definitely of reader interest. But, the length leads to a negative of the manuscript: it is hard to follow the rationale for experimental changes between the parts and hard to follow to follow the conclusions etc. I would think it would be better split up into 2 or more focused manuscripts. In addition, there are many parts that need clarification. I have highlighted many of them, but i fear that there are many more that also need clarification. I would encourage the authors to broaden the comments below into other figures/concepts that similarly need clarification.

Response: We thank the reviewer for this suggestion. We have splitted the contents and in this revised manuscript, we focus on the intrinsic cell-type selectivity of the transcranial focused ultrasound (tFUS), which we believe is most exciting. We have also taken the suggestion and reevaluated unclear sections throughout the revised manuscript.

It would be helpful to have scan tank pictures of the focal spot without the cranium, alongside Fig 2b. It's surprising to me that the rat skull would aberrate the beam enough to give such a weird spot shape in Fig 2b. Fig 2c and 2d are too dark to see the colors.

Response: Figure 2 is updated by changing the previous panel (B) of a coronal view to present a bigger picture of ultrasound beam behind the rat skull with a sagittal view to be consistent regarding the views' orientations presented in physical scanning as panel (C) as well as in computer simulations as panel (D). We also add a new panel (C) to illustrate the focal spot without the cranium alongside Fig. 2B. Figs. 2C and D become Figs. 2D and E, which have been adjusted regarding the color brightness in order to see the colors.

Figure 2. The characterizations of the tFUS pressure fields.

Were there any studies to demonstrate how much ultrasound penetrates when a 40° angle is use on the rat skull? It might be nice to have a pair of figures on this alongside those above.

Response: From Figs. 2B and C, the acoustic insertion loss of ultrasound pressure can be calculated as -5.5 dB. There were previous studies, such as Tufail et al. *Nat. Protoc.*, 2011¹, using angled incidence for tFUS, but no physical measurement was achieved to demonstrate the acoustic insertion loss due to the ~45° incidence angle with the presence of rodent skull.

The text says that intensities are provided in Table 1, but they are not.

Response: Many thanks for catching this missing information in Table 1. In this revision, I_{sppa} , I_{spta} and I_{sptp} are all presented in Table 1, as well as the newly added Table 2 for a constant ultrasound duty cycle.

Table 1. Administered tFUS Conditions with Constant TBD

tFUS Conditions*	UPRF (Hz)	UDC (%)	I_{spta} (mW/cm ²)
UPRF 30Hz	30	0.6	1.05

UPRF 300Hz	300	6	10.55
UPRF 1500Hz	1,500	30	52.74
UPRF 3000Hz	3,000	60	105.48
UPRF 4500Hz	4,500	90	158.22

* Except for the ultrasound pulse repetition frequency (UPRF), the ultrasound duty cycle (UDC), and the spatial-peak temporal-average intensity (I_{spta}), all of the listed tFUS conditions keep the following parameters constant, i.e. ultrasound fundamental frequency (UFF = 500 kHz), ultrasound duration (UD = 67 msec), cycle per pulse number (100), tone-burst duration (TBD = 200 μ sec), spatial-peak pulse-average intensity (I_{sppa} = 175.80 mW/cm²), and spatial-peak temporal-peak intensity (I_{sptp} = 351.61 mW/cm²). Note: the inter-sonication interval (ISol) has a typical value of 2.5 sec and has 10% jittering.

Table 2. Administered tFUS Conditions with Constant UDC of 60%

tFUS Conditions*	UPRF (Hz)	CPP	TBD (msec)
UPRF 30Hz UDC 60%	30	10000	20
UPRF 300Hz UDC 60%	300	1000	2
UPRF 1500Hz UDC 60%	1,500	200	0.4
UPRF 3000Hz UDC 60%	3,000	100	0.2
UPRF 4500Hz UDC 60%	4,500	67	0.134

* Except for the UPRF, CPP and TBD, all of the listed tFUS conditions keep the following parameters constant, i.e. UFF = 500 kHz, UD = 67 msec, I_{spta} = 105.48 mW/cm², I_{sppa} = 175.80 mW/cm², and I_{sptp} = 351.61 mW/cm². Note: the ISol has a typical value of 2.5 sec and has 10% jittering.

When were ketamine injections done?

Response: The ketamine was administered initially with xylazine and was later injected between sessions with toe pinch test. Generally, the additional ketamine was administered roughly every 1 hour in order to extend the anesthesia.

In the calculation for I_{spta} , it's not clear why there is the factor of UPRF. It would seem that the integral over time would have sufficed. Maybe the confusion is because it does not specify the time of the integral.

Response: The calculation for I_{spta} is based on the equation given by US FDA's information documentation on diagnostic ultrasound systems and transducers in 2019, Tufail et al. *Neuron*, 2010², and Tufail et al. *Nat. Protoc.* 2011¹. In those published works, this intensity is defined as $I_{spta} = PII \cdot PRF$, in which the PII is the pulse intensity integral

(equal to the energy fluence per pulse) and the UPRF is indeed a factor for calculating I_{spta} . The reviewer is correct that the confusion may come from the time duration of the integral. Based on the definition and equation for PII given by the US FDA's documentation³, the time duration of this integral operation is the pulse duration (i.e. 1.25 times the interval between the time when the time integral of intensity in an acoustic pulse at a point reaches 10 percent and when it reaches 90 percent of the PII). This is now further clarified in the METHODS section by citing these sources.

Is the claim that the focus is within 1 mm of the skull because that is where the maroon color is? The focus is usually defined as the -3dB line, so that would be much bigger, in the green color.

Response: Yes, our claim is based on the maroon color. Thanks for this comment. We agree with the reviewer's comment on focus, although we make this 1-mm claim for "spatial peak energy", rather than "focus". The sentence is now updated to that "The ultrasound spatial map from the sagittal view (X-Z plane) was reconstructed, in which mechanical energy was distributed along a sagittal beam up to a depth of 5 mm (-3 dB along the axial direction), but spatial peak energy was focused within 1 mm behind the skull (Fig. 2B)."

Why was the sham SSHK not included in Fig 3A?

Response: Thanks for suggestion. The Figure 3A is now updated by adding the exemplified S1 MUA acquired in the sham condition of SSHK (with ultrasound power of 100% and 43%) with UPRF of 1500 and 3000 Hz. All conditions are implemented with isoflurane anesthesia.

Figure 3. The neuronal spiking activities in response to tFUS.

In Fig S2, it doesn't look like there is an increase in spiking in the live animal, like there is in Fig 1A. Why is that?

Response: There is time-locked spiking rate increase in Figure S2D. Thanks for this comment, and we have updated this figure for a better visualization. As seen in Fig S2D, high amplitude neuron spiking activities are only present in the live brain (shown in red), recordings of the dead brain (shown in black) do not exhibit any neural activities. The difference between the live and dead brain illustrates that neural activities can be clearly distinguished from noise or electrical artifacts from tFUS.

Figure S2. Spatial Specificity of tFUS Induced Brain Activations.

Fig 3 shows time locked signals in response to the ultrasound. But, it seems that the data in Fig 4 is not time locked data. Why? Why was time locked data shown in one Figure, then abandoned for the next figure?

Response: Thanks for this comment. Both Figures 3 and 4 are time-locked signals. As it is already explicitly noted in the text body that “In the tFUS group, we compiled all single unit activities during the UD of 67 msec recorded from the rats and separated them into RSUs and FSUs using k-means cluster analysis (Fig. 4A)”, the spiking rate data in Figure 4 were all acquired from the sonication time window. To make this clear in the revision, we updated this description to be that “We compiled all single unit activities during the UD of 67 msec recorded from the rats and separated them into RSUs and FSUs using k-means cluster analysis (Fig. 4). In the tFUS group, 199 identified single units were separated into the two groups, with the RSU group containing 146 units (Fig. 4A).”

Are there 146 dots in each of the bars of Fig 4B? Are there 53 dots in each of the bars in Fig 4C?

Response: Thanks for these questions. The total number of dots in Fig. 4B is 146 for regular spiking units (RSUs), and each bar has shown the sample numbers with those physical dots in the figure. To make this clear, the sample numbers in each of the bars of Fig. 4B are listed in the following table.

Ultrasound Conditions	Number of Samples
UPRF 30Hz	25
UPRF 300Hz	32

UPRF 1500Hz	24
UPRF 3000Hz	31
UPRF 4500Hz	34

The total number of dots in Fig. 4C is 53 for the fast spiking units (FSUs). The sample number of each bar has also shown with physical dots in this figure. The following table describes the number of samples individually for this FSU group.

Ultrasound Conditions	Number of Samples
UPRF 30Hz	5
UPRF 300Hz	7
UPRF 1500Hz	13
UPRF 3000Hz	10
UPRF 4500Hz	18

In this sub-study, we sorted all the spiking single neuronal units with a total amount of 199 from 9 rats and grouped them into RSUs and FSUs for the non-parametric Kruskal–Wallis one-way ANOVA for comparing the multiple groups of different sample sizes. In average, our 32-channel micro-electrode array detected more than 20 single neuronal units, including more than 15 RSUs and more than 5 FSUs from each animal anesthetized with ketamine and xylazine cocktail. The RSU/FSU sample sizes are on par with those being reported by Andermann et al. *Neuron*, 2004⁴, such as their cortical recordings and observations from 13 FSUs and 31 RSUs in one experimental condition and 7 FSUs and 19 RSUs in another condition.

In Fig 4, what is the meaning of the horizontal variation within a UPRF category?

Response: The meaning of the horizontal variation within a UPRF category is indicating the increase of UPRF while listing the UPRFs levels relatively while considering UPRF of 30 Hz as the basis. Within each UPRF category, the X coordinates of the raw data points are plotted among a random distribution in order to lay out all data points. The variation is purely for visual clarity of each data points as the other publications do for data visualization and has no specific meaning. All data points at each UPRF category received the same stimulation. To make this meaning clear, we have changed the labeling for the UPRF category, and have clarified each UPRF value.

Why was the simulation in Fig S1 done at 220 kHz when the rest of the study was done at 500 kHz?

Response: The reason for us to include the 220 kHz simulation in Fig. S1 is to discuss the potential presence of ultrasound standing waves inside a full rat skull structure. As we already discussed that “Although we did not observe significant activations at auditory cortex (Fig. S2A-C) when the tFUS was targeting at S1, the potential auditory confounding factor still needs to be carefully controlled. The section of “Local Activations are Preserved in tFUS When Auditory Pathway is Blocked” in the supplements includes experimental details and results from chemically deafened rats and genetically deafened mice. Here, we provide one of the possible explanations to the previously observed auditory confounding effects, where observations in small rodent models might be confounded by extensive ultrasound standing wave field within the rodent skulls.”, we want to put caution

to the extensive standing wave pattern when a lower ultrasound fundamental frequency is used. The 3D computer simulation results at 500 kHz have already been included in Fig. 2D-E in this revision.

Where is Fig S2 E-F?

Response: Thanks very much for catching this figure labeling mismatch. Fig. S2E should be Fig. S2B, and Fig. S2F should be Fig. S2C. These typos are corrected in this revision.

On page 11, it is not clear why the change in anesthesia was made. It may have changed everything and should have been kept constant.

Response: Thanks for this comment. The reviewer is right as we did introduce two anesthesia methods. As we discussed already in the section of “Study Limitation and Future Investigations” that “In this work, the rodent models were sedated by anesthetic agents, which may introduce an inevitable confounding factor of changing the neuronal spiking activities. In particular, although widely used in ultrasound brain stimulation studies, the injection of ketamine/xylazine does not provide a constant anesthesia level, which is why we also repeated our stimulation under isoflurane anesthesia and normalized to the trial specific baseline.” The reason for us to change the anesthesia is to maintain relatively constant levels of anesthetic effects across experiment trials, which has been mentioned on page 11. Furthermore, since different anesthesia types can lead to different neural excitability and baseline activity, we also wanted to show that our findings persist across different anesthesia types. These two anesthesia methods are widely used in ultrasound neuromodulation literature.

In Fig 5, it is not clear what is meant by Ultrasound was delivered in two grades: 100% and 43%. 43% of what? Why 43%?

Response: 100% SSHK means that we applied the ultrasound field shown in Fig. 2C (without skull) directly onto the electrode shank. In the METHODS, we included brief descriptions about how the 4 levels of ultrasound pressures were identified and applied to the electrode shank. In the sham SSHK conditions, as the setup shown in Fig. 1G and Fig. 5D, the ultrasound was delivered directly onto the middle of electrode shank without penetrating the skull. This is the first difference of SSHK from the tFUS setup shown in Fig. 1A. From Fig. 2B-C, the acoustic pressure insertion loss of the 500 kHz tFUS due to the skull bone is -5.5 dB. This is the first ratio (-5.5 dB = 53% of pressure amplitude decrease) which needs to be derated from the full ultrasound pressure transmitted in the tFUS setup. The second difference between these two setups is the locations on the electrode shank for sonication targeting, i.e. at shank tip in tFUS sonication as shown in Fig. 2D versus at the middle of the shank in the SSHK condition (Fig. 5D). We thus numerically modeled the maximum electrode deflection displacement at the electrode tip using a cantilever model (https://www.engineeringtoolbox.com/cantilever-beams-d_1848.html) when the tFUS wave targeting at the tip or at the middle part. This leads to the second ratio of 81%. We multiplied those two major ratios and obtained 53% x 81% = 43%. This 43% is a rough estimation to derate the full ultrasound pressure being

transmitted in the tFUS setup for the sake of reproducing a roughly equivalent vibration at the electrode tip in both tFUS and SSHK conditions. To make this SSHK sham experiments a rigorous control study in order to eliminate the confounding factor of electrode vibration at the electrode sensing tip, we thus introduced those 4 pressure levels of SSHK, i.e. 100%, 43%, 28% and 13%, in order to cover the full range of possible electrode mechanical vibrations' amplitude due to the tFUS incidence.

To further investigate, a COMSOL model was created to simulate electrode deflection properties while implanted in the brain and an ultrasound induced force is being applied. Those simulation details can be found in the supplementary materials (Fig. S8).

A significant limitation to this study is that the I_{spta} is not kept constant. This should be discussed appropriately in the discussion. As is, it does not really shed light on the effect of duty cycle seen by others, when they do keep I_{spta} constant.

Response: Thanks for this suggestion. In order to keep the I_{spta} constant, we conducted a new set of experiments with all UPRF levels at 60% duty cycle, thus constant I_{spta} . The stimulation parameters are shown in Table 2, and more details are shown in the table below. We delivered the new tFUS parameter set in 7 animals, under isoflurane anesthesia. The number of ultrasound cycles per pulse were changed, which also resulted in changes of TBD, to achieve the same I_{spta} across different UPRFs. As seen in Fig. 6, when stimulated at a constant 60% duty cycle, different UPRFs results in significantly different spiking rates in RSUs (Fig. 6B), while exhibiting no changes in the spiking rate of FSUs (Fig. 6C). (For RSUs: Kruskal-Wallis chi-squared = 70.61, $p = 1.687 \times 10^{-14}$, Fig. 6B; UPRF 30Hz vs. UPRF 300Hz: $p = 4.52 \times 10^{-5}$; UPRF 30Hz vs. UPRF 1500Hz: $p = 5.76 \times 10^{-11}$; UPRF 30Hz vs. UPRF 3000Hz: $p = 4.18 \times 10^{-9}$; UPRF 30Hz vs. UPRF 4500Hz: $p = 2.86 \times 10^{-8}$; UPRF 300Hz vs. UPRF 4500Hz: $p = 1.05 \times 10^{-3}$; UPRF 1500Hz vs. UPRF 3000Hz: $p = 9.30 \times 10^{-5}$, UPRF 1500Hz vs. UPRF 4500Hz: $p = 7.92 \times 10^{-7}$.) Two-way ANOVA testing factors of cell type and UPRF, as well as their interaction show significant interaction between the cell type and UPRF (Fig. 6D). In the same subject population, no significant differences was observed under sham SSKF stimulation.

UPRF (Hz)	Duty cycle	Number of Pulses	Sonication Duration (sec)	Cycles per pulse	Tone-burst duration (μ sec)	Mean trial Duration (sec)
30	60%	2	0.067	10,000	20,000	2.5
300	60%	20	0.067	1,000	2,000	2.5
1500	60%	100	0.067	200	400	2.5
3000	60%	200	0.067	100	200	2.5
4500	60%	300	0.067	67	133	2.5

Figure 6. Validation of cell-type selective responses to UPRF in constant duty cycle.

Two anesthetics were used, but it is not clear why and what is the effect of doing so.

Response: We did use two anesthesia methods. As we discussed already in the section of “Study Limitation and Future Investigations” that “In this work, the rodent models were sedated by anesthetic agents, which may introduce an inevitable confounding factor of changing the neuronal spiking activities. In particular, the injection of ketamine/xylazine does not provide a constant anesthesia level, which is why we also repeated our stimulation under isoflurane anesthesia and normalized to the trial specific baseline.” The reason for us to initially adopt ketamine and xylazine for anesthesia is to keep the same method as used in Tufail et al. *Neuron* 2010 and Tufail et al. *Nat. Protoc.* 2011, and the reason for us to change the anesthesia from ketamine and xylazine cocktail to isoflurane is to maintain relatively constant levels of anesthetic effects across experiment trials, which has been mentioned on page 11. Changing the anesthesia method from ketamine/xylazine to isoflurane allows us to minimize the effect of anesthesia depth. In this revision, we added further statistical analyses, 2-way ANOVA to study the interaction between the cell type and the UPRF. From the statistical results, although we randomized the order of ultrasound sessions, as the reviewer pointed out that such an experimental paradigm may not be able to eliminate/minimize the effect of anesthesia depth by ketamine/xylazine dynamics on neuronal spiking activities during the *in-vivo* neural recordings. In the 2-way ANOVA, the interaction between cell type and UPRF is not significant despite that the UPRF itself plays a significant role in changing the neuronal spiking rate. Therefore, we pursued the isoflurane for a more constant anesthetic solution than the ketamine/xylazine in our later experiments to confirm the cell type specific responses and further examine the interaction between the cell type and UPRF as the two factors. As expected, very significant interactions were found with isoflurane anesthesia (Fig. 5C and Fig. 6D). These new results have been added to the Results.

The manuscript is quite long and I wonder if it wouldn't be better broken up into separate manuscripts. The discussion is quite long and it is not clear that all the points are will supported. It is not clear that the mechanism discussion is supported by the data, at least this is not clearly laid out.

Response: Thanks for this valuable suggestion. We have splitted the contents, and focused only on the intrinsic functional neuron-type selectivity of tFUS in this revised manuscript.

The question about the vibration is not clearly removed when the ultrasound is directed at the shank. It would be nice to have a better picture of the setup. More importantly, it would be nice to have an independent measure of the vibration of the MEA when the US field is directed at the tissue and when directed at the shank, to know if it is a good control.

Response: Thank you for the suggestion. We have spent further efforts to simulate and compare the mechanical vibrations of the MEA both in the tFUS and SSHK conditions. In this simulation, we first numerically calculated the spatial-peak acoustic radiation force (ARF) at a brain volume of e.g. $3 \times 3 \times 3 \text{ mm}^3$ (3 mm is the wavelength of 500 kHz ultrasound at the soft tissue). The 500 kHz acoustic attenuation coefficient (α) of brain tissue is 3 Np/m, the I_{spta} is listed in Table 1 for all the five conditions at different UPRF levels, and the speed of sound (c) in the brain tissue is 1,515 m/s. The estimated ARF magnitude at

the spatial peak volume of 27 mm³ is calculated based on a simplified equation $F = 2 \cdot \alpha \cdot I_{\text{spta}} / c$ and listed below.

Ultrasound Conditions	Spatial Peak ARF magnitude (nN)
UPRF 30Hz	10.9
UPRF 300Hz	109.9
UPRF 1500Hz	549.6
UPRF 3000Hz	1099.2
UPRF 4500Hz	1648.8

Based on the table above, we calculated the amount of ARF experienced by the physical volume of the electrode in the tFUS focus. The electrode (10 mm x 50 μm x 400-125 μm tapered) is modeled with 1 mm of electrode tip in brain tissue (width 125 μm) and 9 mm in air. The electrode end that interfaces in circuitry and housing (width 400 μm) is in air and set as a fixed constrain, as seen in Fig. S8A-B shown below. With the geometric conditions held constant, two scenarios were studied. When tFUS is directed at the tip of the electrode, ARF is applied at the surface of 1 mm segment of electrode inserted into the brain. The applied total ARF magnitude and the calculated displacement are shown in Fig. S8C. When tFUS is directed at the shank of the electrode, ARF is applied at a 3-mm segment of electrode in air centered at the midpoint, approximated by the -3 dB tFUS pressure profile. The applied total ARF magnitude and the calculated displacement are listed in Fig. S8D. The presented simulation ARF magnitude in Fig. S8D accommodates adjustment for pressure difference across the skull, thus closely models that of 43% SSHK condition. Our results estimate the maximum electrode displacement occurs at the tip of the electrode. Moreover, when tFUS is applied at the shank of the electrode in the 43% SSHK, the resulting displacement is greater than typical experimental conditions, i.e. when tFUS is applied at the electrode tip. Therefore, our simulations suggest, 43% SSHK is an adequate sham condition to examine the effect of electrode vibration on neuronal activation.

Figure S8. Numerical Simulations of Microelectrode Array (MEA) As a Single Fixed-edge Cantilever Beam Inserted in Brain.

Reviewer #2 (Remarks to the Author):

This is an interesting study that sought to understand the effect of ultrasound pulse repetition frequency (UPRF) on subtypes of cortical neuronal cells, namely excitatory or inhibitory. Two models were used (1 rat and 1 mouse) to functionally separate cells (Regular-spiking units RSU; excitatory vs Fast-spiking units FSU; inhibitory) or (PV interneurons; inhibitory vs CaMKII-alpha; excitatory). Lastly, an investigation of UPRF on lasting effects such as long-term potentiation (LTP), long-term depression (LTD), and short-term depression (STD) was conducted. The results presented in the study are well described and provide interesting connections to previous studies which may have wide reader interest.

However, the study could benefit from clarification on the novelty and justification of the methodology as well as corrections to critical assertions. Specific comments are as follows:

1. Figure 3 - only two sham conditions were shown whereas 3 sham conditions were introduced in Figure 1. It would be interesting to see the MUA data for the shank sham condition, especially for the same anesthesia condition.

Response: Thanks for the suggestion. In this revision, we include the MUA examples of all 3 sham conditions in the new Figure 3.

Figure 3. The neuronal spiking activities in response to tFUS.

2. Figure 3 C,D,F,G bottom spike plots are hard to see

Response: As the new Figure 3 shown above, we have enhanced the figure contrast for those spike raster plots.

3. Introduction, first para: “designer receptors exclusively activated by designer drugs (DREADDS) etc., have been developed in order to modulate and study the brain” Please replace ‘etc’ with the actual other agents and provide corresponding references.

Response: Thanks for the comment. In this revision, we have replaced the “etc” with more specific brain modulation technologies that “...designer receptors exclusively

activated by designer drugs (DREADDS)^{5,6}, sonogenetics^{7,8}, microbubble-assisted drug delivery⁹ and sonoselective transfection¹⁰, and nanomaterial-mediated magnetic stimulation¹¹⁻¹⁴ have been developed in order to modulate and study the brain.”

4. Page 4, 2nd para: Remove ‘mechanical’ after ‘ultrasonic’

Response: It is removed in the revision accordingly.

5. Page 5, ‘Similar to electrical stimulation, we hypothesize electrical stimulation.’ This hypothesis is not new and several others have already provided comparison of tFUS to electrical with several similarities. Please indicate the exact novelty of this study.

Response: Thanks for the comment. Based on the suggestion by the reviewers, we now split the contents and in this revised manuscript, we focus on the intrinsic cell-type selectivity of the transcranial focused ultrasound (tFUS). The study of sustained synaptic plasticity effects of tFUS will be moved to another manuscript, and we will adjust this statement based on this constructive comment in that manuscript.

6. Page 7, last para: Indicate how artifacts were identified and/or removed at preprocessing.

Response: Thanks for the comment. In this revision, we added more details about artifacts identification and removal in the “MUA and LFP Data Processing” section of “Quantification Analysis” in the Methods. The details are that “For spike analysis, neural traces were band-passed between 244 Hz and 6 kHz. In some of the noisy datasets, the band-pass filtered MUA data were further processed in order to increase the signal-to-noise ratio (SNR). The potential recording artifacts existing in the MUA were identified by computing the inter-spike interval (ISpl) and examining its temporal statistical distribution presented in the return plots (i.e. Poincaré plot, a second-order analysis method for nonlinear features in time series), e.g. Figs. S3B-C and S4B, D, F. In the cases of noisy datasets, Symlet wavelet denoising was applied to all the recording channels using Wavelet toolbox in MATLAB v9.0.0 (The MathWorks, Inc., Natick, MA, USA) to remove the potential artifacts, such as electromagnetic interferences, inefficient ground coupling, and possible mouth movement (e.g. licking). All MUA spike sorting and single-unit preselection are performed using PCA based spike classification software Offline Sorter (Plexon, Dallas, TX, USA). Local field potentials (LFP) were band-passed from 1 Hz to 244 Hz. The band-pass filtered LFP data were then averaged across trials time-locked to the verum/sham ultrasound onset with trial numbers indicated with the descriptions of respective sub-studies. In the case of obvious pre-stimulus fluctuations, the LFP data were further denoised using Wiener filter and independent component analysis in the MATLAB to generate Fig. S10C.”

7. Page 8, paragraph 1, line 4 – Ref 60-62 show RSU/FSU for activation of receptive fields rather than direct activation of the neurons themselves, since, in the present study, tFUS is acting on the neurons directly, how applicable is this metric?

Response: Thanks for this comment. We agree with the reviewer that Ref 60-62 show these two types of neurons in response to stimuli at receptive fields rather than direct activation. In our study, we use the ultrasound to directly interact with RSUs/FSUs, and believe this metric is applicable in this brain stimulation scenario. One example is that McCormick et al. applied this RSU/FSU metric to investigate their different response characteristics to the intracellular electrical current stimulation *in vitro*. So, the activation of receptive fields is not required in order to apply this RSU/FSU metric. In fact, the researchers^{15,16} distinguishing the neurons with this metric is to label the recorded single units to specific neuronal types or functions. In this revision, we also cite the works by McCormick et al.¹⁵ and by Snyder et al.¹⁶ to justify our grouping principle for the cortical neurons.

8. Page 8, paragraph 1, line 9 – Ref 61 contradicts the authors' hypothesis that UPRF drives RSU/FSU through intrinsic cellular differences. Ref 61 states RSU/FSU have distinct responses not because of the intrinsic cellular differences but rather due to differences in the strength of the thalamocortical inputs targeting each cell type. Please address.

Response: Thanks for this comment. Ref 61 by Mountcastle et al. 1969 does look into the three classes of cortical neuronal elements, i.e. the thalamocortical fibers, the thin spikes (discharged by stellate cells) and the regular cortical neurons (largely the pyramidal cells), and their responses to the frequency/amplitude tuning at the receptive field. The reason for us to cite this work is that they proposed specific cortical neuron types corresponding to the RSUs and FSUs. In this specific work, they state that “A major problem in understanding the functional mechanisms of a cortical sensory area is to classify its neurons in terms of **their dynamic and static properties**, with **particular reference** to the first-order afferent fibers of peripheral nerves which project upon them” and they “refer only to convergence among specific thalamocortical afferents, and their linked second- and first-order elements...”. In this particular study, they looked into the serial position of the neurons “in the intracortical inter-neuronal chain leading from input to output”. Their input is the mechanical flutter/vibrational stimuli delivered to the skins of monkeys' hands (also human subjects). Their findings will not contradict our hypothesis regarding that the ultrasound drives RSU/FSU through intrinsic cellular differences. The reason is that their study focused on the different temporal patterns within the discharge trains of those Pacinian cortical neurons activated by the flutter/vibration (up to 300 Hz stimuli at the peripheral receptive fields), whereas in our study, the transmitted tFUS energy majorly interacts with the cortical neurons directly. Nonetheless, we consider this neuronal network and projections as a non-trivial factor effecting on the activities of RSUs and FSUs. In the discussion, we mentioned that “A potential confounding factor to the above experiments is the lack of control for the effects of synaptic transmission between local RSU and FSU neurons of interest and the upstream innervation from surrounding areas. Although unlikely to be present in the current data due to averaging of large number of test trials (e.g. > 400) and short spike response time, the possible effect of cross-cortical or inter-cortical communication cannot be exclusively discounted from the current experiment setup.”

9. Page 10: ‘The baseline firing rate of each single unit in.... tFUS conditions’ Why not monitor anesthesia depth and relate it to the firing rate? Randomizing does not necessarily reduce the anesthesia effect as it also has a dynamic component.

Response: Thanks for this comment. We did record the vital signals, such as the heart rate and breath rate during the MUA recordings. And we noticed the correlation between the baseline firing rate and anesthetic depth. We also found that the baseline firing rate can also be affected by other factors, such as the time duration of the recordings and thus the animal vital state (e.g. dehydration, exhaustion). Considering all these factors, we thus choose the randomization of sessions to mitigate those factors effecting the baseline firing rate. For this reason, we updated this statement as: “The baseline firing rate of each single unit in different ultrasound sessions, which is affected by anesthetic depth with ketamine and xylazine cocktail and time duration of recordings, was mitigated by randomizing the order of ultrasound conditions on each animal subject for rigorous statistical comparisons.” In order to ensure the level of anesthesia not confounding observed results, we repeated our experiment under isoflurane. The continuous gas delivery of isoflurane ensures the consistency of anesthesia depth. In this dataset we observed similar results as the original findings.

10. Page 13, 1st para: Please provide evidence of standing waves. The 500 KHz frequency has a 3 mm wavelength and the mouse brain is typically 7 mm in depth, it is not clear how standing waves would be generated.

Response: Thanks for this comment. The reviewer is right that the mouse brain has a typical depth around 7 mm, which is larger than the 3 mm ultrasound wavelength. But the standing wave is formed mainly due to the interaction between the incidence wave through the cranium and reflected wave from the skull base. The probability of such interactions would be increased mainly due to three factors that 1) the ultrasound incidence approaches a normal direction to the skull, 2) the size of skull cavity gets smaller, and 3) the multiple cycle-per-pulse number used in each ultrasound pulse. The volume of the mouse skull cavity is roughly 5-8 times smaller than that of a Wistar rat skull. The computer simulations of 500 kHz ultrasound for the mouse head model have been implemented using the K-Wave toolbox in this revision (Fig. S1G-H). The extensive ultrasound pressure field inside the full mouse skull originated from the standing wave pattern is observed from the simulation results.

Figure S1G-H. Computer simulations of 500 kHz ultrasound for a mouse head model.

11. Page 13, 2nd para: How was the focusing performed in 'deep brain'?

Response: Thanks for the comment. We discussed the ultrasound focusing performance at the deep brain that “Due to the aperture of coupling collimator limiting the ultrasound to planar wave propagation which further weakens the axial focusing of single-element focused ultrasound transducers, we cannot achieve focal activation only in the deep brain regions. Currently we cannot assert that in activating the deep brain, no other brain regions in the path of the ultrasound beam is activated, this would be achieved in future experiments using phased array focused ultrasound with refocusing techniques¹⁷”. Based on the suggestion by the reviewers, we now split the contents and in this revised manuscript, we focus on the intrinsic cell-type selectivity of the transcranial focused ultrasound. The study of sustained effects of tFUS at the deep brain will be moved to another manuscript, and we will add more descriptions based on this question in that separate manuscript.

12. Page 13: '180 degrees away from the skull' what does this mean?

Response: This description is used to describe the SFLP configuration, and we have depicted this ultrasound setup in Fig. 1E. But to answer the reviewer's question, it means that the active acoustic aperture was flipped upside down, i.e. SFLP configuration. This sham condition completely decouples the tFUS from the skull, but the main transducer body is maintained at the same distance from the rat head in order to control for audible noise and electromagnetic noise generated by the piezoelectric element. The study of the sustained effects of tFUS at the deep brain will be moved to another manuscript, and we will have the updated descriptions based on this question in that separate manuscript.

13. Page 16, paragraph 1, line 4 –data showing that results suggest tFUS affects ion channel dynamics or neuronal morphology is missing.

Response: Thanks for this comment. What we discussed here is the hypothesized reason for the observed unequal responses of functional neuron types *in vivo*. The investigation into these subcellular processes are beyond the scope of this work. The different ion channel dynamics and neuronal morphologies of those neuron types may lead to their different response profiles to the tFUS. To make our discussion appropriate, we have updated the statement as: “The intrinsic different ion channel dynamics¹⁸ and/or neuronal morphologies, such as profiles of dendritic arbors, of the different functional neuron types may be responsible to their unequal responses to the UPRF change. This merits further investigations.”

14. Page 17, paragraph 1, line 2 – Ref 21 does not have to be placed twice within the same sentence

Response: We have updated this sentence which cites the reference only once.

15. Page 18, paragraph 1 - Acoustic Radiation Force was hypothesized to be driven by UPRF, can you show measurements or models of the types of force your parameters produce?

Response: Thanks for the suggestion. We numerically calculated the spatial peak acoustic radiation force (ARF) at a brain volume of e.g. 3x3x3 mm³ (3 mm is the wavelength of 500 kHz ultrasound at the soft tissue). The 500 kHz acoustic attenuation coefficient (α) of brain tissue is 3 Np/m, the I_{spta} is listed in Table 1 for all the five conditions at different UPRF levels, and the speed of sound (c) in the brain tissue is 1,515 m/s. The estimated ARF magnitude at the spatial peak volume of 27 mm³ is calculated based on a simplified equation $F = 2 \cdot \alpha \cdot I_{\text{spta}} / c$ and listed below. This table is also included in the Methods section of this revision.

Ultrasound Conditions	Spatial Peak ARF magnitude (nN)
UPRF 30Hz	10.9
UPRF 300Hz	109.9
UPRF 1500Hz	549.6
UPRF 3000Hz	1099.2
UPRF 4500Hz	1648.8

16. Since the transgenic mice experiments are essential to the study, the figures should be important enough to move to the main text rather than the supplementary.

Response: As suggested by the reviewer, we have moved the contents of transgenic mice from the supplementary to the main text.

17. Page 19: replace 'sonication frequency' with 'pulse repetition frequency'

Response: Thanks for the suggestion. Based on the suggestion by the reviewers, the study of sustained synaptic plasticity effects of tFUS will be moved to a separate manuscript, and we will adjust this term accordingly in that manuscript.

18. Page 20, paragraph 2, line 9 – NMDA receptors in LTD are still being activated. This is essential for moderate calcium influx so that downregulation of AMPA receptors can occur, decreasing sensitivity to glutamate.

Response: Thanks for the suggestion. Based on the suggestion by the reviewers, the study of sustained synaptic plasticity effects of tFUS will be moved to a separate manuscript, and we will update these related discussions accordingly in that manuscript.

19. It is unclear which area (deep brain) is stimulated.

Response: Based on the suggestion by the other reviewers, the study of sustained synaptic plasticity effects of tFUS will be moved to a separate manuscript, and we will describe the stimulated deep brain area accordingly in that manuscript.

20. Page 25, paragraph 1 – Why was live/dead analysis performed *in vitro* rather than *in vivo*?

Response: Thanks for the comment. This is mainly due to the limitation of our current optical imaging setup and its compatibility with an *in vivo* ultrasound stimulation setup. Although we agree that the *in vivo* analysis on cell viability would be ideal, such *in vitro* evaluation would still eligibly serve our purpose for discussing the ultrasound safety, as the histological analyses we already provided in the supplementary cannot directly report the cell viability with a clear color change.

21. Figure 7 – which sham condition is depicted?

Response: The sham condition in Figure 7 is using the setup of SFLP (Fig. 1E). We will make this clear in our updated texts and figures in a separate manuscript.

22. Adjustment for isoflurane vs ketamine/xylazine may be important for other studies, please elaborate.

Response: Thanks for the comment. Changing the anesthesia method from ketamine/xylazine to isoflurane allows us to minimize the effect of anesthesia depth. In this revision, we added further statistical analyses, 2-way ANOVA to study the interaction between the cell type and the UPRF. From the statistical results, although we randomized the order of ultrasound sessions, as the reviewer pointed out that such an experimental paradigm may not be able to eliminate/minimize the effect of anesthesia depth by ketamine/xylazine dynamics on neuronal spiking activities during the *in-vivo* neural recordings. In the 2-way ANOVA, the interaction between cell type and UPRF is not significant despite that the UPRF itself plays a significant role in changing the neuronal spiking rate. Therefore, we pursued the isoflurane for a more constant anesthetic solution than the ketamine/xylazine in our later experiments to confirm the cell type specific responses and further examine the interaction between the cell type and UPRF as the two factors. As expected, very significant interactions were found with isoflurane anesthesia (Fig. 5C and Fig. 6D). These new results have been added to the Results.

Figure 5C. Statistically significant interaction between the cell type and UPRF was observed with constant ultrasound tone burst duration and cycle per pulse.

Figure 6D. Statistically significant interaction between the cell type and UPRF was observed with constant ultrasound duty cycle.

Reviewer #3 (Remarks to the Author):

This study performed several electrophysiological experiments in rats and mice to determine how specific neurons respond to specific ultrasound pulse repetition frequencies (PRF) and duty cycles (DC). The study is positioned timely within a broader quest of several groups to determine the most effective ultrasonic stimulation protocols and the associated mechanisms of action. The study found that the firing rates of specific neuronal pools can be influenced by PRF or DC.

Both PRF and DC are thought to be important factors in ultrasonic neuromodulation. By design, PRF and DC were confounded in this study (Table 1). Therefore, it is not possible to distinguish whether the observed variability in neuronal responses is due to PRF or DC. Consequently, it is difficult if not impossible to devise a model that can explain the effects. This positions the study to be more descriptive rather than generating normative knowledge for the field.

Response: We thank the reviewer for this excellent comment. In order to keep the I_{spta} constant, we conducted a new set of experiments with all UPRF levels at 60% duty cycle. The stimulation parameters are shown in the Table 2 and shown with more details in the table below. We deliver the new tFUS parameter set in 7 animals, under isoflurane anesthesia. The number of ultrasound cycles per pulse were changed, which also resulted in changes of TBD (Fig. 6A), to achieve the same I_{spta} across different UPRFs. As seen in Fig. 6, when stimulated at a constant 60% duty cycle, different UPRFs results in significantly different spiking rates in RSUs (Fig. 6B), while exhibiting no changes in the spiking rate of FSUs (Fig. 6C). (For RSUs: Kruskal-Wallis chi-squared = 70.61, $p = 1.687 \times 10^{-14}$, Fig. 6B; UPRF 30Hz vs. UPRF 300Hz: $p = 4.52 \times 10^{-5}$; UPRF 30Hz vs. UPRF 1500Hz: $p = 5.76 \times 10^{-11}$; UPRF 30Hz vs. UPRF 3000Hz: $p = 4.18 \times 10^{-9}$; UPRF 30Hz vs. UPRF 4500Hz: $p = 2.86 \times 10^{-8}$; UPRF 300Hz vs. UPRF 4500Hz: $p = 1.05 \times 10^{-3}$; UPRF 1500Hz vs. UPRF 3000Hz: $p = 9.30 \times 10^{-5}$, UPRF 1500Hz vs. UPRF 4500Hz: $p = 7.92 \times 10^{-7}$.) Two-way ANOVA testing factors of cell type and UPRF, as well as their interaction show significant interaction ($p = 4.51 \times 10^{-10}$) between the cell type and UPRF (Fig. 6D). In the same subject population, no significant differences were observed under the sham (Fig. 6E-F), i.e. SSKF stimulation.

UPRF (Hz)	Duty cycle	Number of Pulses	Sonication Duration (s)	Cycles per pulse	Tone-burst duration (μ s)	Mean trial Duration (s)
30	60%	2	0.067	10,000	20,000	2.5
300	60%	20	0.067	1,000	2,000	2.5
1500	60%	100	0.067	200	400	2.5
3000	60%	200	0.067	100	200	2.5
4500	60%	300	0.067	67	133	2.5

Figure 6. Validation of cell-type selective responses to UPRF in constant duty cycle.

However, the finding that distinct neuronal cell types respond to certain ultrasound parameters distinctly is still important for the field.

The findings of Fig. S9 D and E are particularly striking, showing a double dissociation of the effects of PRF (or DC) on putative excitatory and inhibitory cells. It is unfortunate that this approach was affected by noise, and so only low PRFs/DCs could be tested. In either case, I suggest to bring this striking finding early on in a future version of the paper, rather than leaving it for the end of the supplement.

Response: Thanks for the favorable comment and suggestion. As suggested, we have moved the contents of transgenic mice from the supplementary to the main text.

The main claim about the influence of PRF/DC on neuronal firing is based on data shown in Fig. 4B, C.

This claim should be supported by a proper, omnibus test rather than the current multiple comparisons.

In particular, this should be tested using a 2-way ANOVA with factors [cell type], [PRF], and their interaction, [cell type]*[PRF]. If there is a statistically significant leverage of PRF/DC on distinct cells type, this would show as the significant [cell type]*[PRF] interaction term in this ANOVA model.

Visually, this interaction does not appear significant in Fig. 4B, C. It does, however, appear significant under the different mode of anesthesia--isoflurane (Fig. 5B, C). To incorporate this, the factor of anesthesia could be included as a third factor in the ANOVA. One would still test for the interaction of [cell type]*[PRF] in this extended model.

Response: Thanks for the suggestion. In this revision, we implemented the 2-way ANOVA with factors of cell type and UPRF, as well as their interaction. As the reviewer predicted, the [cell type]:[UPRF] interaction is not statistically significant ($p > 0.05$) with the data shown in Fig. 4B and C. However, a very significant interaction between the cell type and UPRF was observed ($p = 3.33 \times 10^{-8}$) with the data presented in Fig. 5A and B using isoflurane for anesthesia (Fig. 5C is shown below). As the reviewer suggested, once we incorporate the anesthesia as the third factor of the ANOVA, the three-way interaction term of [cell type]:[UPRF]:[anesthesia method] is statistically significant ($p = 1.94 \times 10^{-9}$) which indicates that the [cell type]:[UPRF] interaction varies across the different anesthesia method, i.e. ketamine/xylazine cocktail or isoflurane. This indeed indicates that the anesthesia method plays an important role in changing the interaction between the cell type and UPRF. Furthermore, we extend this 2-way ANOVA for examining the interaction term when the ultrasound duty cycle is kept constant at 60% (Fig. 6D is shown below). A statistically significant interaction ($p = 4.51 \times 10^{-10}$) was also found.

Figure 5C. Statistically significant interaction between the cell type and UPRF was observed with constant ultrasound tone burst duration and cycle per pulse.

Figure 6D. Statistically significant interaction between the cell type and UPRF was observed with constant ultrasound duty cycle.

Since the LTD effects (Figs. 6 and 7) do not show significant variability by PRF/DC, these data do not contribute to the present framing of the paper and so should likely be presented in a separate paper.

Response: Thanks for this suggestion. We have separated the manuscript and focused only on the intrinsic functional neuron-type selectivity of tFUS in this revision.

Minor suggestions

"UPRFxX" is cryptic. Why not show the specific frequency/duty cycle values, to spare the reader of having to refer to Table 1.

Response: Thanks for the comment. We have revised the naming convention for ultrasound conditions to be more explicit. The updates are made to all the texts and figures.

"ultrasound is delivered to the shank of the electrode upstream" - upstream is generally used in the context of the direction of flow of neural information; something like "at the electrode base" would be more descriptive.

Response: Thanks for the suggestion. We have corrected this wording issue and used "the shank of the electrode base".

typos:

"So far, none have explored"

Response: Thanks for the catching this typo. We have changed these texts to be that "So far, none has explored...".

"These are early evidences"

Response: Thanks for the catching this typo. We have changed the texts accordingly.

References

- 1 Tufail, Y., Yoshihiro, A., Pati, S., Li, M. M. & Tyler, W. J. Ultrasonic neuromodulation by brain stimulation with transcranial ultrasound. *Nat Protoc* **6**, 1453-1470, doi:10.1038/nprot.2011.371 (2011).
- 2 Tufail, Y. *et al.* Transcranial Pulsed Ultrasound Stimulates Intact Brain Circuits. *Neuron* **66**, 681-694, doi:DOI 10.1016/j.neuron.2010.05.008 (2010).
- 3 FDA, U. S. (ed U.S. Dept. Health and Human Services) (Center for Devices and Radiological Health, Rockville, MD, 2019).
- 4 Andermann, M. L., Ritt, J., Neimark, M. A. & Moore, C. I. Neural Correlates of Vibrissa Resonance: Band-Pass and Somatotopic Representation of High-Frequency Stimuli. *Neuron* **42**, 451-463, doi:10.1016/S0896-6273(04)00198-9 (2004).
- 5 Szabrowski, J. O., Lee-Gosselin, A., Lue, B., Malounda, D. & Shapiro, M. G. Acoustically targeted chemogenetics for the non-invasive control of neural circuits. *Nature Biomedical Engineering* **2**, 475-484, doi:10.1038/s41551-018-0258-2 (2018).
- 6 Wang, H., Xie, M., Charpin-El Hamri, G., Ye, H. & Fussenegger, M. Treatment of chronic pain by designer cells controlled by spearmint aromatherapy. *Nature Biomedical Engineering* **2**, 114-123, doi:10.1038/s41551-018-0192-3 (2018).
- 7 Ibsen, S., Tong, A., Schutt, C., Esener, S. & Chalasani, S. H. Sonogenetics is a non-invasive approach to activating neurons in *Caenorhabditis elegans*. *Nature communications* **6**, 8264, doi:10.1038/ncomms9264 (2015).
- 8 Ye, J. *et al.* Ultrasonic Control of Neural Activity through Activation of the Mechanosensitive Channel MscL. *Nano letters* **18**, 4148-4155, doi:10.1021/acs.nanolett.8b00935 (2018).
- 9 Sun, T. *et al.* Closed-loop control of targeted ultrasound drug delivery across the blood-brain/tumor barriers in a rat glioma model. *Proceedings of the National Academy of Sciences of the United States of America* **114**, E10281-E10290, doi:10.1073/pnas.1713328114 (2017).
- 10 Gorick, C. M. *et al.* Sonoselective transfection of cerebral vasculature without blood-brain barrier disruption. *Proceedings of the National Academy of Sciences of the United States of America* **117**, 5644-5654, doi:10.1073/pnas.1914595117 (2020).
- 11 Dobson, J. Remote control of cellular behaviour with magnetic nanoparticles. *Nat Nanotechnol* **3**, 139-143, doi:10.1038/nnano.2008.39 (2008).
- 12 Rao, S. *et al.* Remotely controlled chemomagnetic modulation of targeted neural circuits. *Nat Nanotechnol*, doi:10.1038/s41565-019-0521-z (2019).
- 13 Huang, H., Delikanli, S., Zeng, H., Ferkey, D. M. & Pralle, A. Remote control of ion channels and neurons through magnetic-field heating of nanoparticles. *Nat Nanotechnol* **5**, 602-606, doi:10.1038/nnano.2010.125 (2010).
- 14 Chen, R., Romero, G., Christiansen, M. G., Mohr, A. & Anikeeva, P. Wireless magnetothermal deep brain stimulation. *Science* **347**, 1477-1480, doi:10.1126/science.1261821 (2015).
- 15 McCormick, D. A., Connors, B. W., Lighthall, J. W. & Prince, D. A. Comparative electrophysiology of pyramidal and sparsely spiny stellate neurons of the

- neocortex. *Journal of neurophysiology* **54**, 782-806, doi:10.1152/jn.1985.54.4.782 (1985).
- 16 Snyder, A. C., Morais, M. J. & Smith, M. A. Dynamics of excitatory and inhibitory networks are differentially altered by selective attention. *Journal of neurophysiology* **116**, 1807-1820, doi:10.1152/jn.00343.2016 (2016).
- 17 Ballard, J. R., Casper, A. J., Wan, Y. & Ebbini, E. S. Adaptive transthoracic refocusing of dual-mode ultrasound arrays. *IEEE Trans Biomed Eng* **57**, 93-102, doi:10.1109/TBME.2009.2028150 (2010).
- 18 Yoo, S., Mittelstein, D. R., Hurt, R., Lacroix, J. & Shapiro, M. G. Focused ultrasound excites neurons via mechanosensitive calcium accumulation and ion channel amplification. *bioRxiv* (2020).

Reviewer #1 (Remarks to the Author):

The manuscript is generally improved, although there are still points that need further clarification.

The introduction is longer than necessary. Specifically, the first paragraph can be eliminated. The second paragraph can be reduced.

There are standard ways to refer to many of the parameters and making new acronyms is unnecessary and confusing. Please use PRF, not UPRF. Please use f_0 (with the 0 being subscripted), not UFF. Please use D, not UD. Please use DC, not UDC. Please use ISI (inter-stimulus interval, rather than iSol).

The subscripts in ISPTP are capitalized, as are ISPTA etc.

It is not clear why there is a blanket statement that all studies were done under IACUC approval, and then this is repeated over and over. The blanket statement should be sufficient.

Specifying the CPP isn't needed if the TBD is provided.

It is not clear what order the figures are referenced.

Keep tense the same. Either all present tense or all past tense.

In Eq. 3, shouldn't there also be a factor of 2 in the denominator from the time average of the \sin^2 ?

What is the reference for the mapping from HU to acoustic velocity in Table 3? It is surprising that a velocity of 3000 m/s was used for rat skull.

How is the range of attenuation values for rat skull used in Table 3? It is surprising that the rat skull has such a large range. What is the reference for this mapping?

The jittering is not explained.

The caption of Fig 4 should make it clear that the Table 1 parameters are used. Same for Fig 5. A plot like that in Fig 6a would be useful in Fig 4.

In Fig 4A, it looks like the number of RSU and FSU neurons is about the same. But there are clearly more data points in Fig 4B than in Fig 4C. Why is that?

Is the lack of significance in Fig 4C just due to the lack of numbers? How was it determined that enough sampling had been done?

Is Fig 5 again using Table 1 parameters?

The text around Fig 5 suggests that anesthesia was studied, but where is the data in Fig 5 about anesthesia?

In the Discussion about the Legon paper, it is not clear what transition zone is being referred to.

This manuscript didn't show any inhibitory effects.

Why was a CW condition not used?

Reviewer #2 (Remarks to the Author):

The authors did an excellent job at addressing all comments including novelty by focusing their paper on the type of neurons modulated by tFUS.

Reviewer #3 (Remarks to the Author):

All of my comments were addressed satisfactorily.

I have a minor suggestion - the "normalized spiking rate" plots may at a first glance be interpreted as firing rates in [Hz]; especially given that this ordinate was used in the previous figures. From that perspective, the effects would be tiny. I suggest to change 1 -> 100%, 2 -> 200%, etc., so that it becomes apparent that the effects are in fact appreciable.

Response to Reviewers

We are extremely grateful to the reviewers for their favorable and constructive comments to our first revision. We have spent efforts to further revise the manuscript in order to address all the reviewers' additional comments.

Our responses to the reviewer comments are shown below in BLUE, the reviewers' original comments are shown in BLACK. Fig. x and Fig. Sx refer to the figures in the main manuscript and in the supplementary materials, respectively. All revised or added texts in the manuscript are with track changes.

Reviewer #1:

The manuscript is generally improved, although there are still points that need further clarification.

The introduction is longer than necessary. Specifically, the first paragraph can be eliminated. The second paragraph can be reduced.

Response: We thank the reviewer for this suggestion. As suggested, we have shortened the first and second paragraphs.

There are standard ways to refer to many of the parameters and making new acronyms is unnecessary and confusing. Please use PRF, not UPRF. Please use f_0 (with the 0 being subscripted), not UFF. Please use D, not UD. Please use DC, not UDC. Please use ISI (inter-stimulus interval, rather than iSol).

Response: As suggested, we have changed the acronyms to PRF, f_0 and DC. We maintained the notation UD for ultrasound duration in order to minimize confusions between ultrasound duration and ultrasound element diameters or aperture size of an ultrasound probe¹, sometimes denoted as D. Also, we maintained the notation ISol for the inter-sonication interval, which is used to differentiate from another notation ISpl for the inter-spike interval used in generating Figs. S3 and S4. Moreover, as we applied the optical stimulation for generating the results presented in Fig. 7, to differentiate the ultrasound stimulation from the optical stimulation in the acronyms, we think ISol might be more explicit to the readers than using the general inter-stimulus interval.

The subscripts in ISPTP are capitalized, as are ISPTA etc.

Response: We thank the reviewer for catching this issue. We have capitalized their subscripts accordingly.

It is not clear why there is a blanket statement that all studies were done under IACUC approval, and then this is repeated over and over. The blanket statement should be sufficient.

Response: The statement is not simply repeated in the Methods. Each of the three animal populations were regulated by a different combination of the two affiliated universities. The first appearance of the IACUC approval is regarding the **rat** studies overseen by **both the University of Minnesota and Carnegie Mellon University**; the second appearance of the IACUC approval is regarding the **optogenetic mice** study overseen by the **University of Minnesota**; the third appearance of this statement is for the **genetically deafened mice** study overseen by **Carnegie Mellon University**. We want to make these regulation related statement clear to the public, and

therefore such statement looks repeated three times. In this revision, we removed the fourth appearance of such statement in order to reduce the redundancy as suggested by the reviewer.

Specifying the CPP isn't needed if the TBD is provided.

Response: As suggested, most of the CPP values were removed from the manuscript, while we only keep one CPP value in the notes of Table 1, and two CPP values in the captions of Fig. 6A.

It is not clear what order the figures are referenced.

Response: Thanks for the suggestion. In this revision, we have spent further efforts to update the referencing texts in the main manuscript as well as change the order of supplementary figures in order to reference all the figures as sequentially as possible.

Keep tense the same. Either all present tense or all past tense.

Response: We thank the reviewer for this suggestion about our writing. We have updated the tense according to the effective writing guidelines by the Scitable (Nature Education).

In Eq. 3, shouldn't there also be a factor of 2 in the denominator from the time average of the \sin^2 ?

Response: Based on the literature²⁻⁵, Equation (3) is in its complete form and there should not be a factor of 2 in the denominator.

What is the reference for the mapping from HU to acoustic velocity in Table 3? It is surprising that a velocity of 3000 m/s was used for rat skull.

Response: The references were already included in the manuscript. Based on literatures, the sound speed in the cortical bone was 3476 m/sec according to Culjat et al.⁶, and this speed was 3100 m/sec according to Mueller et al.⁷. If one wants to describe this acoustic velocity specifically in the rat skull, the 3000 m/sec was used for numerical simulations by Younan et al. 2013⁸, and the 3100 m/sec was used in the computer simulations by Constans et al. 2017⁹.

How is the range of attenuation values for rat skull used in Table 3? It is surprising that the rat skull has such a large range. What is the reference for this mapping?

Response: The reference was included in the manuscript by mentioning that "The simulation study was following a similar protocol described by Mueller et al.⁷ and using the acoustic parameters listed in Table 3." In this reference, the range of attenuation coefficient values of the rat skull is [21.5, 208.9] Np/MHz/m, and once this range gets converted to dB/MHz/cm, one will reach the numerical range, i.e. [1.87, 18.15] dB/MHz/cm as listed in Table 3 in the Methods. Additionally, Constans et al.⁹ used this coefficient as 8 dB/MHz/cm for their simulation, and Younan et al.⁸ used 6.9 dB/MHz/cm for this attenuation coefficient in their simulation study.

The jittering is not explained.

Response: Thank you. The jittering is now explained in the revised Methods. Essentially, this 10% jittering of the inter-stimulus interval is to minimize the timing effect and potential brain adaptation to the tFUS. The inter-sonication interval was randomly selected from a uniform distribution centered at 2.5 sec, bounded by ± 250 msec.

The caption of Fig 4 should make it clear that the Table 1 parameters are used. Same for Fig 5. A plot like that in Fig 6a would be useful in Fig 4.

Response: Thank you for this constructive comment. In this revision, the captions of Figs. 4 and 5 include the statement that “The ultrasound parameters listed in Table 1 were used in this study.” Actually, the suggested plot has already been included as Fig. 1B.

In Fig 4A, it looks like the number of RSU and FSU neurons is about the same. But there are clearly more data points in Fig 4B than in Fig 4C. Why is that?

Response: The sample numbers of RSU and FSU in Fig. 4A looked about the same, but actually, there were multiple data points being overlapped more in the RSU group than in the FSU group. In other words, one data point depicted in the panel A may represent more than one single unit. To illustrate this hidden information, the actual data point histograms for RSUs and FSUs in Fig. 4A are depicted in the figures below. The data point overlapping happened because of the time resolution of the recorded action potentials. This time resolution (0.05 msec) is limited by the sampling frequency of our neural recording devices, i.e. 20 kHz for the Smartbox.

Is the lack of significance in Fig 4C just due to the lack of numbers? How was it determined that enough sampling had been done?

Response: It is possible that the FSUs' lack of significant response to the ultrasound PRF change was due to the limited sample size, i.e. 53. The ketamine/xylazine anesthesia method in this specific study did give us some challenge to record and identify sufficient number of FSUs.

Although this FSUs' sample size was not ideal, the RSU/FSU sample sizes were still on par with those being reported by Andermann et al. *Neuron*, 2004¹⁰, such as their cortical recordings and observations from 13 FSUs and 31 RSUs in one experimental condition and 7 FSUs and 19 RSUs in another condition. Further, given this fact, we changed the anesthesia method to isoflurane in the follow-up studies as presented in Figs. 5 and 6. In these studies, we were able to record more than double of the sample size for FSUs. Similar observations were acquired regarding the homogeneous responses of FSUs to the ultrasound PRF change. Therefore, we conclude that sampling was not the limiting factor for the lack of significance.

Is Fig 5 again using Table 1 parameters?

Response: Yes, it is. In this revision, the caption of Fig. 5 includes the statement that “The ultrasound parameters listed in Table 1 were used in this study.”

The text around Fig 5 suggests that anesthesia was studied, but where is the data in Fig 5 about anesthesia?

Response: All data presented in Fig. 5 were acquired from the rats anesthetized with isoflurane. In this revision, a brief statement that “The isoflurane was used to anesthetize the rats” is included in the caption. Regarding the anesthesia, we did report the results from a 3-way ANOVA test, in which the anesthesia method is considered as a factor in comparing the data presented in Fig. 4A-C using ketamine/xylazine cocktail (data were further normalized in order to be further tested) and the data presented in Fig. 5A-B using isoflurane. In this revision, we included one more figure in the supplementary document, i.e. Fig. S5I (also shown below) to demonstrate the RSUs' spiking rate changes under the isoflurane anesthesia vs. under the ketamine/xylazine cocktail. It can be seen that the RSUs' activities were affected by the anesthesia method significantly. We also updated the main texts with that “Moreover, once the anesthesia method was incorporated as the third factor for the ANOVA, the three-way interaction among cell type, PRF levels, and anesthesia method (i.e., using ketamine/xylazine cocktail with normalizing the data presented in Fig. 4A-C or isoflurane with the normalized data presented in Fig. 5A-B) is statistically significant ($p = 1.94 \times 10^{-9}$). This indicates that the cell type and PRF interaction varies significantly across different anesthesia methods. Another 2-way ANOVA studying how the RSUs' activities were impacted by both the PRF levels and anesthesia methods was illustrated in Fig. S5I. Significant effects of the PRF ($p = 2.02 \times 10^{-6}$) and the anesthesia method ($p = 0.0084$) were found.”

In the Discussion about the Legon paper, it is not clear what transition zone is being referred to. This manuscript didn't show any inhibitory effects.

Response: Thanks for this question. The reviewer is right that our current work did not show any inhibitory effects. When we discussed Legon's paper, the term "transition zone" referred to the results from a computer simulation paper by Plaksin et al. 2016¹¹. In Plaksin et al. 2016, tFUS parameters were classified based on its likelihood to elicit excitatory or inhibitory responses. This paper was cited already in the sentence that "The inhibitory phenomena observed by Legon et al.^{12,13} resulted from a UDC located in a transition zone between tFUS induction of inhibitory and excitatory effects¹¹, whereas the brain activation reported by Lee et al.^{14,15} is likely due to the applied UPRF-related higher UDC." (In this revision, UDC and UPRF have been changed to DC and PRF according to an early suggestion by the reviewer.)

Why was a CW condition not used?

Response: Thanks for the excellent question. Actually, we did introduce the continuous wave mode briefly when we applied the tFUS condition using the PRF 30 Hz and DC 60% listed as in Table 2. The tone-burst duration of this specific ultrasound condition was 20 msec, which is longer than the 10-msec TBD employed in the CW excitation by O'Reilly et al.¹⁶. To avoid the potential confusion about ultrasound mode, in this revision, we revised the statement at the beginning of "Ultrasound Safety" in the Discussion section by removing the word "pulsed ultrasound" to be that "All tFUS stimulation parameters used on the S1 cortices of rats and mice were maintained in brief exposures (i.e. 67-msec sonication per trial, with the duty cycle of each trial being less than 3%) and with low intensities,..."

Reviewer #2:

The authors did an excellent job at addressing all comments including novelty by focusing their paper on the type of neurons modulated by tFUS.

Response: We really appreciate all of the reviewer's constructive questions and favorable comments.

Reviewer #3:

All of my comments were addressed satisfactorily.

I have a minor suggestion - the "normalized spiking rate" plots may at a first glance be interpreted as firing rates in [Hz]; especially given that this ordinate was used in the previous figures. From that perspective, the effects would be tiny. I suggest to change 1 -> 100%, 2 -> 200%, etc., so that it becomes apparent that the effects are in fact appreciable.

Response: We want to thank the reviewer for constructive comments and helpful suggestion. In this revision, we have updated the presentation of numbers for all the normalized spiking rate as suggested by the reviewer.

References:

- 1 Sun, C., Dai, F., Jiang, S. & Liu, Y. in *IEEE Int Ultra Sym.* 1-4.
- 2 Tufail, Y. *et al.* Transcranial Pulsed Ultrasound Stimulates Intact Brain Circuits. *Neuron* **66**, 681-694, doi:DOI 10.1016/j.neuron.2010.05.008 (2010).
- 3 Tufail, Y., Yoshihiro, A., Pati, S., Li, M. M. & Tyler, W. J. Ultrasonic neuromodulation by brain stimulation with transcranial ultrasound. *Nat Protoc* **6**, 1453-1470, doi:10.1038/nprot.2011.371 (2011).
- 4 Ye, P. P., Brown, J. R. & Pauly, K. B. Frequency Dependence of Ultrasound Neurostimulation in the Mouse Brain. *Ultrasound in medicine & biology* **42**, 1512-1530, doi:10.1016/j.ultrasmedbio.2016.02.012 (2016).
- 5 Mohammadjavadi, M. *et al.* Elimination of peripheral auditory pathway activation does not affect motor responses from ultrasound neuromodulation. *Brain Stimulation*, doi:<https://doi.org/10.1016/j.brs.2019.03.005> (2019).
- 6 Culjat, M. O., Goldenberg, D., Tewari, P. & Singh, R. S. A review of tissue substitutes for ultrasound imaging. *Ultrasound in medicine & biology* **36**, 861-873, doi:10.1016/j.ultrasmedbio.2010.02.012 (2010).
- 7 Mueller, J. K., Ai, L., Bansal, P. & Legon, W. Numerical evaluation of the skull for human neuromodulation with transcranial focused ultrasound. *J Neural Eng*, doi:10.1088/1741-2552/aa843e (2017).
- 8 Younan, Y. *et al.* Influence of the pressure field distribution in transcranial ultrasonic neurostimulation. *Medical physics* **40**, 082902, doi:10.1118/1.4812423 (2013).
- 9 Constans, C., Deffieux, T., Pouget, P., Tanter, M. & Aubry, J. F. A 200-1380-kHz Quadrifrequency Focused Ultrasound Transducer for Neurostimulation in Rodents and Primates: Transcranial In Vitro Calibration and Numerical Study of the Influence of Skull Cavity. *IEEE transactions on ultrasonics, ferroelectrics, and frequency control* **64**, 717-724, doi:10.1109/TUFFC.2017.2651648 (2017).
- 10 Andermann, M. L., Ritt, J., Neimark, M. A. & Moore, C. I. Neural Correlates of Vibrissa Resonance: Band-Pass and Somatotopic Representation of High-Frequency Stimuli. *Neuron* **42**, 451-463, doi:10.1016/S0896-6273(04)00198-9 (2004).
- 11 Plaksin, M., Kimmel, E. & Shoham, S. Cell-Type-Selective Effects of Intramembrane Cavitation as a Unifying Theoretical Framework for Ultrasonic Neuromodulation. *eNeuro* **3**, doi:10.1523/ENEURO.0136-15.2016 (2016).
- 12 Legon, W. *et al.* Transcranial focused ultrasound modulates the activity of primary somatosensory cortex in humans. *Nat Neurosci* **17**, 322-329, doi:10.1038/nn.3620 (2014).
- 13 Legon, W., Ai, L., Bansal, P. & Mueller, J. K. Neuromodulation with single-element transcranial focused ultrasound in human thalamus. *Human brain mapping*, doi:10.1002/hbm.23981 (2018).
- 14 Lee, W., Chung, Y. A., Jung, Y., Song, I. U. & Yoo, S. S. Simultaneous acoustic stimulation of human primary and secondary somatosensory cortices using transcranial focused ultrasound. *BMC Neurosci* **17**, 68, doi:10.1186/s12868-016-0303-6 (2016).
- 15 Lee, W. *et al.* Transcranial focused ultrasound stimulation of human primary visual cortex. *Sci Rep* **6**, 34026, doi:10.1038/srep34026 (2016).
- 16 O'Reilly, M. A., Huang, Y. & Hynynen, K. The impact of standing wave effects on transcranial focused ultrasound disruption of the blood-brain barrier in a rat model. *Physics in medicine and biology* **55**, 5251-5267, doi:10.1088/0031-9155/55/18/001 (2010).

Reviewer #1 (Remarks to the Author):

Thank you for your many edits and responsiveness to review. I have a couple more questions here about the responses.

>>What is the referencne for the mapping from HU to acoustic velocity in Table 3? It is surprising that a velocity of 3000 m/s was used for rat skull.

>Response: The references were already included in the manuscript. Based on literatures, the sound speed in the cortical bone was 3476 m/sec according to Culjat et al.6, and this speed was 3100 m/sec according to Mueller et al.7. If one wants to describe this acoustic velocity specifically in the rat skull, the 3000 m/sec was used for numerical simulations by Younan et al. 20138, and the 3100 m/sec was used in the computer simulations by Constans et al. 20179.

Mueller et al 85 is in human, not rat.

Culjat doesn't say anything about rat.

Younan does do rat simulations, but they picked their numbers based on Marquet that did it in humans.

Constans is not clear how they got their numbers for rat skull.

I didn't go through the attenuation references, but I suspect they are equally unclear.

I realize that this is a minor point for this manuscript, but I suggest the authors think carefully about their references, rather than cite lots of references that don't make sense. If it were me, I might only cite one of the Aubry papers, which at least uses these values for rat. It is surprising that there aren't better literature values for rat.

>Therefore, we conclude that sampling was not the limiting factor for the lack of significance.

I'm not really convinced that the lack of points in 4c isn't what leads to lack of significance.

>>Why was a CW condition not used?

>Response: Thanks for the excellent question. Actually, we did introduce the continuous wave mode briefly when we applied the tFUS condition using the PRF 30 Hz and DC 60% listed as in Table 2. The tone-burst duration of this specific ultrasound condition was 20 msec, which is longer than the 10-msec TBD employed in the CW excitation by O'Reilly et al.16. To avoid the potential confusion about ultrasound mode, in this revision, we revised the statement at the beginning of "Ultrasound Safety" in the Discussion section by removing the word "pulsed ultrasound" to be that "All tFUS stimulation parameters used on the S1 cortices of rats and mice were maintained in brief exposures (i.e. 67-msec sonication per trial, with the duty cycle of each trial being less than 3%) and with low intensities,..."

Not clear - are you saying that CW and 60% DC are the same?

Are you saying that all studies were done with <3% DC, even though Table 2 is 60% DC?

Response to Reviewer Comments

We are very grateful to the reviewer for the additional constructive comments to our second revision. We have considered very carefully on each of the reviewer's comments and spent further efforts to revise the manuscript in order to address all the additional comments.

Our responses to the reviewer comments are shown below in **BLUE**, and the reviewer's original comments are shown in **BLACK**. Fig. x refers to the figures in the main manuscript. All revised or added texts in the manuscript are printed in **RED**.

We hope this revision has satisfactorily addressed the reviewer's comments.

Reviewer #1:

>>What is the referencne for the mapping from HU to acoustic velocity in Table 3? It is surprising that a velocity of 3000 m/s was used for rat skull.

>Response: The references were already included in the manuscript. Based on literatures, the sound speed in the cortical bone was 3476 m/sec according to Culjat et al.6, and this speed was 3100 m/sec according to Mueller et al.7. If one wants to describe this acoustic velocity specifically in the rat skull, the 3000 m/sec was used for numerical simulations by Younan et al. 20138, and the 3100 m/sec was used in the computer simulations by Constans et al. 20179.

Mueller et al 85 is in human, not rat.

Culjat doesn't say anything about rat.

Younan does do rat simulations, but they picked their numbers based on Marquet that did it in humans.

Constans is not clear how they got their numbers for rat skull.

I didn't go through the attenuation references, but I suspect they are equally unclear.

I realize that this is a minor point for this manuscript, but I suggest the authors think carefully about their references, rather than cite lots of references that don't make sense. If it were me, I might only cite one of the Aubry papers, which at least uses these values for rat. It is surprising that there aren't better literature values for rat.

Response: We thank the reviewer very much for this excellent point. First of all, to be more rigorous about the specific type of bone, we unified our terms "cranial bone" in line 732 on page 36 and in Table 3 and "cortical bone" in line 731 on page 36 and in line 761 on page 37 to be "cortical bone".

We have carefully reviewed various relevant literatures and agree with the reviewer in regard to the sound speed value of rat skull. As the reviewer pointed out, there are no better rat-specific values for this parameter among the literature. Many transcranial ultrasound studies on rat models adopted the acoustic properties, such as the speed of sound [1-3] and the ultrasound attenuation coefficient of bone [3, 4] from the human/non-rat skull values with the assumption that such values would not differ significantly among animal/human skull models.

Younan et al. [3] performed computer simulations on rat model using the sound speed value (3000 m/sec) described in Marquet et al.'s work [5] which studied on monkey and human skull models. Actually, Marquet et al. [5] briefly mentioned "the skull's high acoustic speed of sound (about 3000 m s⁻¹)" in the introduction section, but did not use the 3000 m/sec for the skull's sound speed. Instead, Marquet et al. [5] used a more precise value on page 2603 for the cortical bone, which was 3100 m/sec for their study. (The bone's density of 2200 kg/m³ was also described on this page in [5]).

For this reason, we chose the 3100 m/sec for the rat cortical bone, and cited Constans et al. [6], in which the sound speed value of 3100 m/sec was introduced to their simulations on monkey models, and they also did extensive simulations on rat models in this work.

To justify our choice of the 3100 m/sec for the rat cortical bone, we numerically estimated the range of the speed of sound in the cortical bone of rats based on the following equation [7] for solids (source: The Physics Hypertextbook): $v = \sqrt{\frac{E}{\rho}}$, in which E is the Young's modulus (elastic modulus) and ρ is the medium density. Cory et al. [8] reported comprehensive measurements of compressive mechanical properties of rat bone, in which the elastic modulus of the rat cortical bone was measured as 18.98±4.78 GPa, and the medium density of the rat cortical bone was measured as 2167±17 kg/m³. By plugging these numbers to the above equation, we obtained a range of the speed of sound for the rat cortical bone, i.e., 2550 – 3320 m/sec, and the mean value was 2960 m/sec. If we plug the human values, i.e., E = 18.6±3.5 GPa [7] and $\rho = 1908\pm133$ kg/m³ (source: Density » IT'IS Foundation), we can obtain a range of the speed of sound for the human cortical bone, i.e., 2720 – 3530 m/sec, and the mean value is 3120 m/sec (similar to the sound speed values being used by Mueller et al. [9] and Marquet et al. [5]). Therefore, it can be seen from the above numerical calculations that the speeds of sound in rat and human cortical bones are close.

To be concise, as the reviewer suggested, we now only cite one Aubry paper, i.e. Constans et al. [6] in this revision and assume the speed of sound at 3100 m/sec for the cortical bone in our simulations. We adopted the acoustic parameters in Table 3, followed the similar modeling protocol as in Mueller et al. [9], and solved the equations (1) – (4) in [9] based on our own rodent CT data in order to obtain the whole skull-wide spatial distribution of skull density, speed of sound and acoustic attenuation.

We have made the following revisions:

On page 36, lines 730-732, we updated the texts from “Where ρ is the medium density (1,028 kg/m³ for brain tissue, 1,975 kg/m³ for cortical bone⁸²), and c is the speed of sound in the medium (1,515 m/sec for brain tissue, 3100 m/sec for cranial bone⁸³).“ to that “Where ρ is the medium density (1028 kg/m³ for brain tissue, assuming 2200 kg/m³ for cortical bone⁸⁴), and c is the speed of sound in the medium (1515 m/sec for brain tissue, assuming 3100 m/sec for cortical bone⁸⁵).”

On page 36, line 744-746, we further clarified the method for the mapping from HU to the acoustic properties by updating the descriptions from “The simulation study was following a similar protocol described by Mueller et al.⁸⁵ and using the acoustic parameters listed in Table 3” to that “The simulation study was following a similar protocol described by Mueller et al.⁸⁷ in order to obtain the spatial distributions of medium density, speed of sound and acoustic attenuation throughout the rodent skull based on the porosity calculated from CT Hounsfield units⁸⁸ using the assumed acoustic parameters of cortical bone listed in Table 3.”

>Therefore, we conclude that sampling was not the limiting factor for the lack of significance. I'm not really convinced that the lack of points in 4c isn't what leads to lack of significance.

Response: We thank the reviewer for raising this point. We agree with the reviewer that for the specific statistical testing in Fig. 4C, the lack of points may lead to the lack of significance. Actually, we did state in the beginning of our response to the reviewer's comment (in our last revision), that “It is possible that the FSUs' lack of significant response to the ultrasound PRF change was due to the limited sample size, i.e. 53.” We also agree with the reviewer that our statement, “Therefore, we conclude that sampling was not the limiting factor for the lack of significance.” in the previous response letter is not rigorous. To address this sampling size issue, we changed the anesthesia method to isoflurane in the follow-up studies as presented in Figs. 5 and 6. In these studies, we were able to record more than double of the sample size for FSUs. Similar observations were acquired regarding the homogeneous responses of FSUs to the ultrasound PRF change. Our intended statement is: “Therefore, given the consistent observations in Figs. 4B-C, 5 and 6, we believe that the different responses observed from RSUs vs. FSUs to various PRF levels are not due to the sample numbers.” We have also discussed this issue in the **Study Limitation and Future Investigations** of the **Discussion** section as follows:

On page 23-24, lines 495-503, we added the texts that “Note that with the ketamine/xylazine anesthesia, it is possible that the FSUs’ lack of significant response to the ultrasound PRF change was due to the limited sample size (Fig. 4C). To address this sampling size issue, we changed the anesthesia method to isoflurane in the follow-up studies (Figs. 5 and 6). In these studies, we were able to record from more FSUs. Similar observations were acquired regarding the lack of significant responses of FSUs to different levels of ultrasound PRF. Given the consistent observations in Figs. 4B-C, 5 and 6, we believe that the different responses observed from RSUs vs. FSUs to various PRF levels are not due to the sample numbers.”

>>Why was a CW condition not used?

>Response: Thanks for the excellent question. Actually, we did introduce the continuous wave mode briefly when we applied the tFUS condition using the PRF 30 Hz and DC 60% listed as in Table 2. The tone-burst duration of this specific ultrasound condition was 20 msec, which is longer than the 10-msec TBD employed in the CW excitation by O’Reilly et al.16. To avoid the potential confusion about ultrasound mode, in this revision, we revised the statement at the beginning of “Ultrasound Safety” in the Discussion section by removing the word “pulsed ultrasound” to be that “All tFUS stimulation parameters used on the S1 cortices of rats and mice were maintained in brief exposures (i.e. 67-msec sonication per trial, with the duty cycle of each trial being less than 3%) and with low intensities,…”

Not clear - are you saying that CW and 60% DC are the same?

Response: Thank you very much for this comment. We assume that the reviewer might consider the CW as 100% DC during the 67-msec ultrasound duration (UD). The reason for us not investigating this CW is that we aim to investigate the neuronal effect of the ultrasound PRF in the present work, which requires us to repeat the ultrasound pulse at certain frequency. In other words, the 100% DC of CW mode does not allow us to investigate this important ultrasound parameter. Furthermore, the 67-msec or longer CW stimulation may lead to increased local temperature rise which may introduce confounding factors that can lead to changes in neural excitability [10, 11]. Nevertheless, we agree with the reviewer that it would be interesting to study how the RSUs and FSUs would respond to the CW ultrasound. Therefore, we added a brief discussion in this revision on page 25-26, lines 539-543 in **Study Limitation and Future Investigations** that “A future study would be helpful by substantially characterizing how the continuous wave (CW) ultrasound configuration would impact on the intrinsic neuron-type selectivity of the tFUS, while the temperature rise due to the CW needs to be carefully controlled and minimized in order to avoid local temperature changes, which are known modulators of neural excitability^{76,77}.”

We would also like to clarify that in our previous statement we did not intend to imply that the CW is equivalent to the 60% DC. We previously considered our stimulation at PRF 30 Hz 60% DC, which consists of 2 long tone burst duration (TBD) of 20 msec within the 67-msec ultrasound duration, similar to what some literatures in the field claims as CW. For example, in a very recent publication by Lu et al. [12], “two 2-ms-long CWs following a rest of 6 s. The interval between two waves was 20 ms” were applied to stimulate the visual cortex of rats. In another particular CW excitation profile by O’Reilly et al. [13], they applied “continuous wave (CW) excitation delivered in 10 ms bursts at a repetition rate of 1 Hz”. Given the long TBD used in our PRF 30 Hz 60% DC, we felt as if, the result from this specific condition may shed light on the effect of CW stimulation in our system. We apologize for the confusion.

Are you saying that all studies were done with <3% DC, even though Table 2 is 60% DC?

Response: Thank you for raising this point. The term “duty cycle” is confusing in this case because it can describe the ratio between the ultrasound duration vs. trial duration, as well as the ratio between the TBD vs. pulse repetition period (1/PRF). When we described “the duty cycle of each trial being less than 3%”, we referred to “the duty cycle over the whole experiment, accounting for the ISI between bursts” [14], also known as total duty cycle (TDC, defined in [14]). TDC is calculated from the ratio of ultrasound duration 67msec to the inter-sonication interval which was 2.5 sec±10% jittering in our experiments. In contrast, the DC defined in our manuscript is the duty cycle within the ultrasound duration (UD, i.e. 67 msec), known as burst duty cycle (BDC, defined in [14]). We are sorry that we did not make this point clear in our previous

revision. In this revision, we updated the statement to be that “All tFUS stimulation parameters used on the S1 cortices of rats and mice were maintained in brief exposures (i.e. 67-msec sonication per trial, with the **total** duty cycle of each trial being less than 3%) and with low intensities,...” (page 22), and we have further clarified the definition of our DC in the Methods section that “ultrasound duty cycle (DC, within the ultrasound duration of 67 msec)” (page 29).

The following revisions have been made:

On page 22, line 473, we specified the “duty cycle” as “total duty cycle”.

On page 29, line 602, we specified the DC as “within the ultrasound duration of 67 msec”.

References:

- [1] M. A. O'Reilly, A. Muller, and K. Hynynen, "Ultrasound Insertion Loss of Rat Parietal Bone Appears to Be Proportional to Animal Mass at Submegahertz Frequencies," *Ultrasound in Medicine and Biology*, vol. 37, no. 11, pp. 1930-1937, 2011, doi: 10.1016/j.ultrasmedbio.2011.08.001.
- [2] M. Gerstenmayer, B. Fellah, R. Magnin, E. Selingue, and B. Larrat, "Acoustic Transmission Factor through the Rat Skull as a Function of Body Mass, Frequency and Position," *Ultrasound in Medicine and Biology*, vol. 44, no. 11, pp. 2336-2344, 2018, doi: 10.1016/j.ultrasmedbio.2018.06.005.
- [3] Y. Younan, T. Deffieux, B. Larrat, M. Fink, M. Tanter, and J. F. Aubry, "Influence of the pressure field distribution in transcranial ultrasonic neurostimulation," *Medical physics*, vol. 40, no. 8, p. 082902, Aug 2013, doi: 10.1118/1.4812423.
- [4] P. C. Tsai, H. S. Gougheri, and M. Kiani, "Skull Impact on the Ultrasound Beam Profile of Transcranial Focused Ultrasound Stimulation," *Annu Int Conf IEEE Eng Med Biol Soc*, vol. 2019, pp. 5188-5191, Jul 2019, doi: 10.1109/EMBC.2019.8857269.
- [5] F. Marquet *et al.*, "Non-invasive transcranial ultrasound therapy based on a 3D CT scan: protocol validation and in vitro results," *Physics in medicine and biology*, vol. 54, no. 9, pp. 2597-613, May 7 2009, doi: 10.1088/0031-9155/54/9/001.
- [6] C. Constans, T. Deffieux, P. Pouget, M. Tanter, and J. F. Aubry, "A 200-1380-kHz Quadrifrequency Focused Ultrasound Transducer for Neurostimulation in Rodents and Primates: Transcranial In Vitro Calibration and Numerical Study of the Influence of Skull Cavity," *IEEE transactions on ultrasonics, ferroelectrics, and frequency control*, vol. 64, no. 4, pp. 717-724, Apr 2017, doi: 10.1109/TUFFC.2017.2651648.
- [7] J. Y. Rho, R. B. Ashman, and C. H. Turner, "Young's modulus of trabecular and cortical bone material: Ultrasonic and microtensile measurements," *Journal of biomechanics*, vol. 26, no. 2, pp. 111-119, 1993/02/01/ 1993, doi: [https://doi.org/10.1016/0021-9290\(93\)90042-D](https://doi.org/10.1016/0021-9290(93)90042-D).
- [8] E. Cory, A. Nazarian, V. Entezari, V. Vartanians, R. Müller, and B. D. Snyder, "Compressive axial mechanical properties of rat bone as functions of bone volume fraction, apparent density and micro-ct based mineral density," (in eng), *Journal of biomechanics*, vol. 43, no. 5, pp. 953-960, 2010, doi: 10.1016/j.jbiomech.2009.10.047.
- [9] J. K. Mueller, L. Ai, P. Bansal, and W. Legon, "Numerical evaluation of the skull for human neuromodulation with transcranial focused ultrasound," *J Neural Eng*, Aug 04 2017, doi: 10.1088/1741-2552/aa843e.
- [10] K. Shibasaki *et al.*, "TRPV4 activation at the physiological temperature is a critical determinant of neuronal excitability and behavior," *Pflügers Archiv - European Journal of Physiology*, vol. 467, no. 12, pp. 2495-2507, 2015/12/01 2015, doi: 10.1007/s00424-015-1726-0.
- [11] K. Shibasaki, M. Suzuki, A. Mizuno, and M. Tominaga, "Effects of body temperature on neural activity in the hippocampus: regulation of resting membrane potentials by transient receptor potential vanilloid 4," *The Journal of neuroscience : the official journal of the Society for Neuroscience*, vol. 27, no. 7, pp. 1566-75, Feb 14 2007, doi: 10.1523/JNEUROSCI.4284-06.2007.
- [12] G. Lu *et al.*, "Transcranial Focused Ultrasound for Noninvasive Neuromodulation of the Visual Cortex," *IEEE transactions on ultrasonics, ferroelectrics, and frequency control*, vol. 68, no. 1, pp. 21-28, Jan 2021, doi: 10.1109/TUFFC.2020.3005670.
- [13] M. A. O'Reilly, Y. Huang, and K. Hynynen, "The impact of standing wave effects on transcranial focused ultrasound disruption of the blood-brain barrier in a rat model," *Physics in medicine and biology*, vol. 55, no. 18, pp. 5251-67, Sep 21 2010, doi: 10.1088/0031-9155/55/18/001.
- [14] J. Blackmore, S. Shrivastava, J. Sallet, C. R. Butler, and R. O. Cleveland, "Ultrasound Neuromodulation: A Review of Results, Mechanisms and Safety," *Ultrasound in medicine & biology*, May 18 2019, doi: 10.1016/j.ultrasmedbio.2018.12.015.